# LLE-MORL: Locally Linear Extrapolation of Policies for Efficient Multi-Objective Reinforcement Learning

## Abstract

Multi-objective reinforcement learning (MORL) aims at optimising several, often conflicting goals in order to improve the flexibility and reliability of RL in practical tasks. This can be achieved by finding diverse policies that are optimal for some objective preferences and non-dominated by optimal policies for other preferences so that they form a Pareto front in the multi-objective performance space. The relation between the multi-objective performance space and the parameter space that represents the policies is generally non-unique, and we provide new insights into this by formalising a local parameter-performance relationship. Using a training scheme based on the local parameter-performance relationship, we propose LLE-MORL, a method that directly extrapolates a small set of base policies to efficiently trace out a high-quality Pareto front. Experiments conducted with and without retraining across different domains show that LLE-MORL consistently achieves higher Pareto front quality and efficiency than state-of-the-art approaches.

## 1 Introduction

Reinforcement Learning (RL) has shown great promise in complex decision-making problems, enabling significant advancements in a wide range (Silver et al., 2016; Levine et al., 2016). In real-world scenarios, however, problems often feature multiple, often conflicting, objectives. Under this circumstance, multi-objective approaches provide flexibility in practical applications of reinforcement learning by providing a modifiable policy that can be adjusted according to changes of preference among a set of objectives (Roijers et al., 2013; Hayes et al., 2022). This has fostered the development of the field known as multi-objective reinforcement learning (MORL). Ideally, the modifiable policies developed within MORL allow efficient adaptation, ensuring that a policy optimal for one set of preferences can be readily transformed to be optimal for a new set when those preferences change. To prepare such a modifiable policy for application, three problems have to be solved: (i) The *learning* problem involves the solution of an RL problem for each combination of preference parameters or at least for a representative subset of preferences. (ii) The *representation* problem requires a parametrization of the policies, which typically results in either a discrete set of individual policies (common in population-based methods) or a single, continuously adaptable policy (prevalent in deep reinforcement learning approaches). (iii) The *selection* problem is to identify a suitable policy in the application which includes dynamic adjustments to preference drifts and possibly the decision whether a different policy should be invoked or whether further training is required to respond to a temporary detection of suboptimality.

We propose to consider these problems as a coherent task, in order to reduce the computational burden of the learning problem and improve the interpretability of the policy representation. We hypothesise that if a continuous representation of policies can be found where similar preferences correspond to similar policy parameters, then small performance differences might be compensable with brief, targeted retraining. It is also anticipated that such a structured and interpretable policy representation would benefit the selection problem, though this aspect is not the primary focus of our current study.

While a globally continuous mapping is an ideal, we notice that in non-trivial problems, the relationship between the performance space and parameter space of policies is not a simple, single continuous mapping but can be described by a family of locally continuous components (Xu et al., 2020; Li et al.,

2024). Our findings suggest that effectively exploring just a few of these components can be sufficient to achieve competitive performance in typical benchmark problems. This understanding forms the basis of our core concept: the Parameter-Performance Relationship (PPR). The process is seeded by an RL task that finds a good but not necessarily optimal parameter vector for an initial policy. Then a second policy is obtained by retraining with different preferences, and from there, additional policies are efficiently generated by a locally linear extrapolation, which led to the name LLE-MORL for the approach that we present in the following. If the policies obtained by the extension process are briefly retrained, they can improve with further extension, although eventually they may become dominated by earlier solutions which would indicate the need for a restart with a different initial policy. Within each solution component, the policy representation is easily interpretable in terms of the continuous PPR, but also the boundaries where the policies depart from optimality are interesting. They indicate that a discontinuous reparametrisation takes place and that thus policies of a potentially qualitatively different type are optimal on either side of the boundary.

Building upon the notions of PPR and locally linear extension, in this paper, we introduce LLE-MORL, a MORL algorithm that is designed to efficiently trace the Pareto front (see Sect. 2.2) by systematically exploring these identified local structures. Our experiments demonstrate that the proposed algorithm can achieve high-quality Pareto front approximations with notable sample efficiency. This strong performance is primarily attributed to its simple yet effective locally linear extension mechanism, which significantly reduces the need for extensive retraining along the Pareto front. Such efficiency is made by exploiting the locally continuous nature of the parameter-performance relationship, a characteristic that also enhances the overall interpretability of our approach.

## 2 BACKGROUND

### 2.1 MULTI-OBJECTIVE REINFORCEMENT LEARNING

Multi-Objective Reinforcement Learning (MORL) extends the traditional RL framework to scenarios where agents must consider multiple, often conflicting objectives. This extension allows for more sophisticated decision-making models that mirror real-world complexities where trade-offs between competing goals, such as cost versus quality or speed versus safety, are common. To ground this notion formally, we represent a MORL problem as a Multi-Objective Markov Decision Process (MOMDP) which generalises the standard MDP framework to accommodate multiple reward functions, each corresponding to a different objective.

**Definition 1.** *Multi-Objective Markov Decision Process (MOMDP). A MOMDP is defined by the tuple $(\mathcal{S}, \mathcal{A}, \mathcal{P}, \{\mathcal{R}^d\}, \gamma, \Omega, f_\Omega)$, where $\mathcal{S}$ is the state space, $\mathcal{A}$ is the action space, $\mathcal{P}(s'|s,a)$ is the state transition probability, $\mathcal{R}^d$ is a vector-valued reward function with $d$ as the number of objectives, specifying the immediate reward for each of the considered objectives, $\gamma$ is the discount factor, $\Omega$ is the preferences space, $f_\Omega : \mathbb{R}^d \to \mathbb{R}$ is the scalarisation function.*

The crucial difference between MOMDPs and traditional single-objective MDPs is the reward structure. While single-objective MDPs use a scalar reward function $\mathcal{R}$, MOMDPs feature a vector-valued reward function $\mathcal{R}^d$ that delivers distinct numeric feedback for each objective, directly correlating the length of the reward vector with the number of objectives. At each timestep $t$, the agent in state $s_t \in \mathcal{S}$ selects an action $a_t \sim \pi(\cdot \mid s_t)$, transitions to a new state $s_{t+1}$ with probability $P(s_{t+1} \mid s_t, a_t)$, and receives a reward vector $\mathbf{r}_t = \big[(R_1(s_t, a_t), R_2(s_t, a_t), \ldots, R_d(s_t, a_t)]\big)$. We define the discounted return vector by $\mathbf{G}_t = \sum_{k=0}^{\infty} \gamma^k \mathbf{r}_{t+k}$, and the multi-objective action-value function of a policy $\pi$ for a given state-action pair $(s, a)$ by $\mathbf{Q}^\pi(s, a) = \mathbb{E}_\pi[\mathbf{G}_t \mid s_t = s, a_t = a]$. The goal of MORL is to find a policy $\pi$ such that the expected return of each objective can be optimised. In practice, we trade off objectives via a scalarisation function $f_\omega(\mathbf{r})$, which produces a scalar utility using preference vector $\omega \in \Omega$. The scalarisation function $f_\omega(\mathbf{r})$ is used for mapping the multi-objective reward vector $\mathbf{r}(s, a)$ to a single scalar. In this paper, we consider the linear scalarisation function $f_\omega(\mathbf{r}(s, a)) = \omega^{\mathbf{T}} \mathbf{r}(s, a)$, which is commonly used in MORL literature (Yang et al., 2019; Felten et al., 2024). When the preference dimension $d = 1$ (so that the return vector is one-dimensional), the MOMDP collapses to a standard single-objective MDP, since the reward vector reduces to a scalar and $f_\Omega$ becomes the identity mapping.

## 2.2 PARETO OPTIMALITY

In multi-objective optimisation, the concept of optimality differs from the single-objective case. Typically, no single policy simultaneously maximises all objectives, due to inherent trade-offs. Without any additional information about the user's preference, there can now be multiple possibly optimal solutions. In the following, we introduce several useful definitions for possibly optimal policies.

**Definition 2.** *Pareto optimality. A policy $\pi$ is said to* dominate *another policy $\pi'$ if and only if $\forall i \in \{1, \ldots, d\}$, $V_i^\pi(s) \geq V_i^{\pi'}(s)$, and $\exists j, V_j^\pi(s) > V_j^{\pi'}(s)$, where $V_i^\pi(s) = \mathbb{E}_\pi[\sum_{t=0}^\infty \gamma^t R_i(s_t, a_t) \mid s_0 = s]$ denotes the expected discounted return for objective $i$ under policy $\pi$. A policy $\pi^*$ is* Pareto optimal *if and only if it is not dominated by another policy. The set of all Pareto optimal policies forms the* Pareto set*: $\mathcal{P} = \{\pi \mid \pi \text{ is Pareto optimal}\}$. The corresponding set of expected returns incured by policies in the Pareto set is termed* Pareto front*: $\mathcal{F} = \{V^\pi(s) \mid \pi \in \mathcal{P}\}$.*

Since obtaining the true Pareto set is intractable in complex problems, the practical aim of multi-objective optimisation is to construct a finite set of policies that closely approximate the true Pareto front. So that practitioners can select the policy based on their preferred trade-off among objectives.

## 3 METHODS

### 3.1 OVERVIEW

As shown in Figure 1, we first explore the relationship between the parameter space and the performance space. We empirically find that a short retraining of a converged policy under a new preference induces a small, structured update in parameter space that corresponds to a predictable shift of the expected returns of the policy. This "model similarity"—the fact that the retrained policy stays close to the original parameters while already moving toward a different region of the Pareto front—underpins our method for steering policies along the front. Building on this insight, we explore the possibility that using the parameter-space difference between two structurally similar policies—trained under different preferences—to guide directional updates that extend our approximation of the Pareto front.

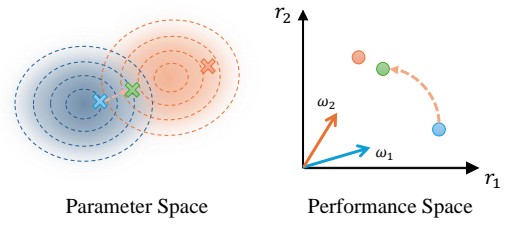

Figure 1: Parameter space and performance space of the Pareto policies. (Left) 2D projection of the high-dimensional policy parameter space. Red and blue gradient shadings and contour lines depict the scalarised reward under different preference vectors $\omega_1$ and $\omega_2$. The arrow marks a short retraining update. (Right) The policy obtained by retraining the $\theta_{w_1}$ model under $\omega_2$ (green) shifts towards the new preference as seen in performance space.

Leveraging this property, we develop an efficient algorithm to approximate the Pareto set of policies. We start by initializing a small collection of base policies, each trained to converge under a distinct scalarization weight chosen to span the preference evenly. Next, for each base policy, we perform a short retraining under a different preference weight, capturing the small parameter update that shifts the policy toward a new trade-off. These updates serve as directional moving vectors: we move from each base policy along its vector by a tunable step size to generate intermediate policies. Finally, we apply a brief fine-tuning to each intermediate policy under its corresponding preference, i.e. the scalarization weight shifted by the same fraction as the parameter updated, nudging it onto the true Pareto front.

### 3.2 PARAMETER-PERFORMANCE RELATIONSHIP

Recent work in multi-objective reinforcement learning has implicitly suggested a relationship between the parameter space of the policy network and the Pareto front in the performance space. (Xu et al., 2020) empirically show for PGMORL that each disjoint policy family occupies a continuous region in parameter space and maps to a contiguous segment of the Pareto front, while MORL/D (Felten

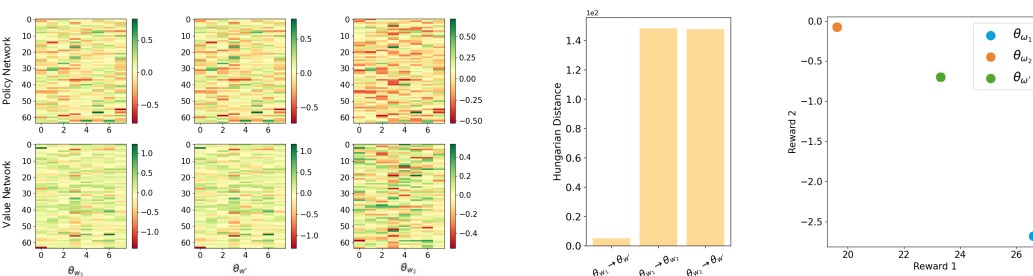

(a) Policy-net and value-net parameter heatmap to illustrate the retraining effect that is measured in (b).

(b) Combined Hungarian model distance.

(c) Respective positions in performance space.

Figure 2: Comparing independently trained policy $\theta_{w_2}$ versus retrained policy $\theta_{w'}$ based on $\theta_{w_1}$, for details see Section 3.3. The environment used here is the multi-objective SWIMMER problem.

et al., 2024) assume that policies with similar parameters should lead to close evaluations. Motivated by these implicit observations, we introduce the a *parameter–performance relationship* and proceed to explain and empirically validate this property.

**Definition 3.** *Parameter-Performance Relationship (PPR). Let $\Theta \subseteq \mathbb{R}^n$ be the policy parameter space and $V : \Theta \to \mathbb{R}^d$ the mapping from parameter vectors $\theta$ to the expected return vectors $V(\theta)$. We say $V$ exhibits a* continuous parameter–performance relationship *on a region $U \subseteq \Theta$ if there exists a function $h : \mathbb{R}^n \to \mathbb{R}^d$ and a radius $\delta > 0$ such that, for any $\theta \in U$ and any parameter perturbation $\Delta\theta$ with $\|\Delta\theta\| < \delta$ and $\theta + \Delta\theta \in U$, $V(\theta + \Delta\theta) - V(\theta) = h(\Delta\theta)$.*

To study this relationship, we first need a metric for policy closeness in parameter space. We adopt the Hungarian matching distance (Kuhn, 1955; Munkres, 1957) to measure model distance and thereby quantify structural similarity between policies. A formal definition of Hungarian matching distance is provided in Appendix C.3. This metric naturally handles the permutation invariance of hidden units (Goodfellow et al., 2016) and measures the smallest structural change needed to align one model to another—lower Hungarian distance indicates greater model similarity.

### 3.3 SANITY CHECK

To get a first idea about the PPR, we compare policies trained independently with those obtained by short retraining. We first train two policies to convergence using a multi-objective PPO-based (Schulman et al., 2017) algorithm with scalarization vectors $w_1$ and $w_2$, yielding model parameters $\theta_{w_1}$ and $\theta_{w_2}$. Starting from $\theta_{w_1}$, we then perform one short additional training step with $w_2$ to obtain $\theta_{w'}$. To quantify how "close" these policy variants are, we show neuron heatmaps for each model both at the policy-network and value-network level in Figure 2a, and visualise the Hungarian matching distances between those models in Figure 2b. We also plot the rewards for three policies in the two-objective performance space (Figure 2c) for the multi-objective SWIMMER problem.

We compare three pairs of models: (1) $\theta_{w_1}$ and $\theta_{w_2}$, capturing differences between independently trained policies in both parameter space and performance space; (2) $\theta_{w_1}$ and $\theta_{w'}$, showing that brief retraining yields a structurally similar model and a low Hungarian matching distance, yet already shifted toward $w_2$ in reward space; and (3) $\theta_{w'}$ and $\theta_{w_2}$, illustrating that although their parameters remain distinct, their rewards lie much closer on the performance space.

These empirical observations show that a short retraining step under a new preference produces a small, structured parameter update that directly maps to a predictable shift in performance, validating the local PPR. More details of the sanity check procedure are provided in Appendix C.4.

### 3.4 LOCALLY LINEAR EXTENSION

Based on the PPR definition, a natural question is whether the parameter-space difference between two structurally similar policies—trained under different preferences—can serve as a directional

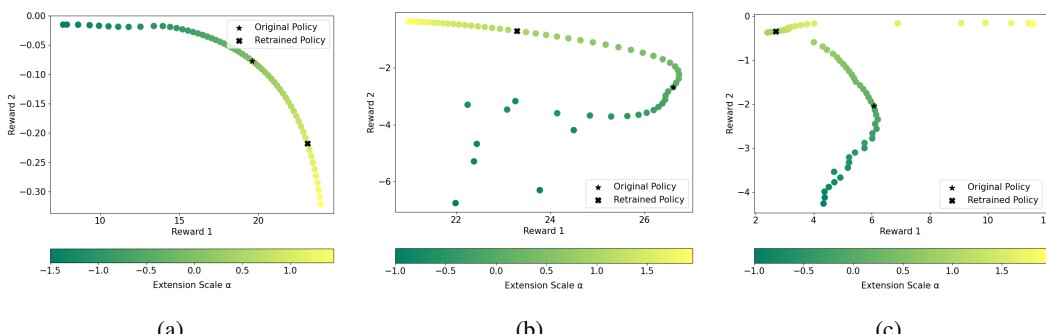

(a)                                    (b)                                    (c)

Figure 3: Visualisation of the process of applying the parameter difference $\Delta\theta = \theta_{w'} - \theta_w$ between two related policies. The policies are obtained by first training a policy $\theta_w$ to convergence using scalarization vector $w$ and then find policy $\theta_{w'}$ by a brief additional training period with a different scalarization vector $w'$. Iterating the shift $\Delta\theta$ in the policy space induces a sequence of shifts also in the multi-objective reward space. The subfigures show results for different initial preferences: (a) A convex front is found from the two policies. (b) Although the original policy turns out to be Pareto suboptimal, the solution manifold extends into a Pareto optimal component. (c) Retraining can cause the (Pareto-suboptimal) original solution to jump to a different branch so that the corresponding solution consists of two components one of which can be ignored because of Pareto suboptimality.

update to extend our approximate Pareto front. To explore this, we consider two policies, a base policy $\theta_w$ and a retrained policy $\theta_{w'}$, which exhibits a parameter-performance relationship. Crucially, for this directional information to be meaningful for Pareto front exploration, both $\theta_w$ and $\theta_{w'}$ should ideally be non-dominated solutions, at least with respect to each other. Given such a pair, we compute the parameter update vector $\Delta\theta = \theta_{w'} - \theta_w$ and generate a set of intermediate policies by moving from the base policy $\theta_w$ along the parameter displacement $\Delta\theta$ in scaled steps. Concretely, for each scale $\alpha$, we form $\theta_\alpha = \theta_w + \alpha\,\Delta\theta$ and evaluate its reward vectors in preference space.

Figure 3 visualises the resulting trajectory of reward vectors in the two-dimensional objective space: as $\alpha$ grows, the trajectory passes through the region around $\theta_{w'}$ and can extend beyond both the base and retrained endpoints, demonstrating how simple parameter-space moves can traverse broad trade-off regions, which offers a cost-effective strategy for efficiently expanding an approximate Pareto front without training each point from scratch.

### 3.5 THE LLE-MORL ALGORITHM

Building on the locally linear extension mechanism, which is critical for tracing an approximate Pareto front, we develop the full algorithm, LLE-MORL (Locally Linear Extrapolation for Multi-Objective Reinforcement Learning). The full Algorithm 1 (see Appendix A) consists of five stages: **(1) Initialisation:** We train a set of $K$ base policies $\{\theta_{w_i}\}_{i=1}^{K}$ to convergence using PPO (Schulman et al., 2017). Each policy is trained under a distinct scalarization weight $w_i \in \Omega$, where these weights are chosen to be evenly distributed across the preference space. **(2) Directional Retraining:** For each $i = 1, \ldots, K-1$, continue train based on $\theta_{w_i}$ under a new preference $w_{i'}$ for $T_{\text{dir}}$ steps to obtain $\theta_{w'}$, where $\theta_{w_i}$ and $\theta_{w_{i'}}$ should be both non-dominated points. Record the parameter update vector $\Delta\theta_i = \theta_{w_{i'}} - \theta_{w_i}$ and weight shift $\Delta w_i = w_{i+1} - w_i$. **(3) Locally Linear Extension:** For each base policy $\theta_{w_i}$, we generate a set of intermediate policies by applying each step-scale factor $\alpha_j$ to the parameter update vector $\Delta\theta_i$. Concretely, each candidate is $\theta_{i,j} = \theta_{w_i} + \alpha_j\,\Delta\theta_i$, allowing negative and positive moves along the local direction in parameter space. Simultaneously, we adjust the preference weight by $\Delta w_i$ scaled by $\alpha_j$ to obtain $w_{i,j}$. These step-scale factors control how far along the local direction each intermediate policy moves. **(4) Candidate Selection:** All candidate policies $\theta_{i,j}$ generated in the locally linear extension stage are evaluated to obtain their respective performance vectors. From this set of extended policies, we identify and select the subset of non-dominated solutions. These selected non-dominated candidates are then advanced to the fine-tuning stage. **(5) Preference-Aligned Fine-Tuning:** from each candidate $\theta$ and its matched weight $w$, perform a short PPO fine-tuning of $T_{\text{ref}}$ steps under $w$ to push the generated policy closer to the true Pareto front.

To clarify when LLE-MORL can reliably reconstruct a Pareto front, we present the theoretical analysis here. We consider here the Pareto fronts $P$ that are differentiable manifolds, meaning that it is locally similar to Euclidean space and derivatives can be defined. As a slight generalisation, we can assume that the Pareto front is a non-connected manifold or a set of manifolds, including zero-dimensional manifolds (discrete points) or sets of manifolds of different dimensions. We will not further discuss this generalisation apart from mentioning the fact that it is possible to reach all components and to produce sufficiently many charts of each of the components of the Pareto front.

For the theorem below, we rely on the following assumptions:

- A1: $P$ is a (set of) differentiable manifold(s) with known dimension(s).
- A2: $P$ does not contain any non-trivial cluster points (limit points).
- A3: The function $h$ that maps from one tangential space to the tangential space of another point that is within a distance controlled by a sufficiently small $\Delta\alpha$ is close to the identity.
- A4: The initialisation (step 1 above) identifies $K$ different points (in the respective component) of the Pareto front.

**Theorem.** *If part of the Pareto front in a $n$-dimensional preference space can represented by a $(n-1)$-dimensional manifold in the policy parameter space, and the parameters of $n$ different nearby solutions on the Pareto front are known, then there exists a $\Delta\alpha > 0$, so that Algorithm 1 (LLE-MORL) reconstructs this part of the Pareto front up to a resolution determined by $\Delta\alpha$.*

Under the conditions of the theorem, we describe the LLE-MORL time complexity as follows.

**Proposition.** *Time complexity. Given that the number of objectives is $n$, the number of base policies is $K$, the base policy training time is $T$, the number of locally linear extensions sampled per direction is $M$, and noting that the locally linear extension is training-free, the expected running time of LLE-MORL is $O(TK + KM^{n-1})$.*

See Appendix B for proofs and further discussions.

## 4 EXPERIMENTS

### 4.1 EXPERIMENT SETUP

In this section, we evaluate the LLE-MORL algorithm using popular continuous MORL benchmark problems from the MO-Gymnasium (Felten et al., 2023). Our benchmark problems include three two-objective continuous environments: **MO-Swimmer-v5**, **MO-Hopper-2d-v5**, **MO-Ant-2d-v5**. We evaluate the quality of the approximate Pareto front using three standard metrics: **Hypervolume (HV)**, **Expected Utility (EU)**, **Sparsity (SP)**, following the formalism in (Zitzler & Thiele, 2002; Zintgraf et al., 2015; Hayes et al., 2022). Higher HV and EU values indicate better overall front quality, while SP measures the distribution of solutions along the front. Since SP is scale-dependent and less directly related to decision quality, we report it mainly as a complementary metric to assess coverage diversity. More details can be found in Appendix C.

We compare our LLE-MORL against the following state-of-the-art MORL algorithms: (i) **GPI-LS** (Alegre et al., 2023) applies Generalised Policy Improvement over a discretised set of preference weights and uses linear scalarization to construct a diverse Pareto set. (ii) Concave-augmented Pareto Q-learning (**CAPQL**) (Lu et al., 2023): learns an ensemble of Q-functions under different preferences and selects actions via conservative aggregation to improve front coverage. (iii) **Q-Pensieve** (Hung et al., 2023) boosts the sample efficiency of MORL by storing past Q-function snapshots in a replay buffer, enabling explicit policy-level knowledge sharing across training iterations. (iv) **MORL/D** (Felten et al., 2024) is a deep-RL analogue of decomposition-based multi-objective optimisation that trains subpolicies under scalarised objectives and recombines them via weight decompositions to approximate the Pareto front.

### 4.2 RESULTS AND ANALYSIS

To assess the performance of MORL, we now present quantitative results evaluating the quality of the approximated Pareto fronts. We conduct experiments under two distinct settings to provide a

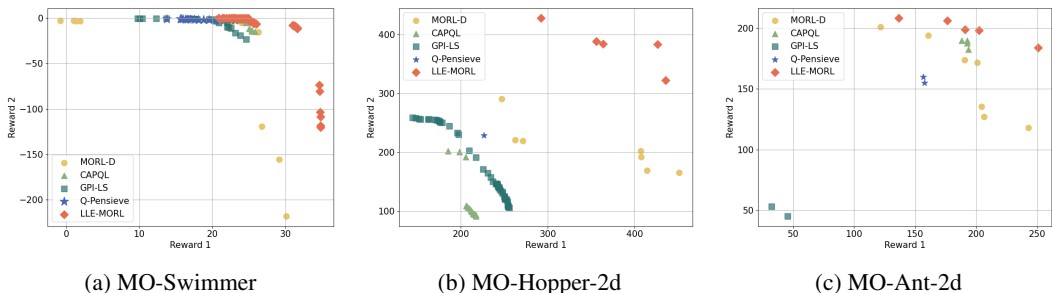

(a) MO-Swimmer  (b) MO-Hopper-2d  (c) MO-Ant-2d

Figure 4: Pareto fronts from the sample-efficient training setting, comparing our LLE-MORL method with baselines on three continuous-control benchmarks. LLE-MORL demonstrates more comprehensive Pareto fronts across all benchmarks.

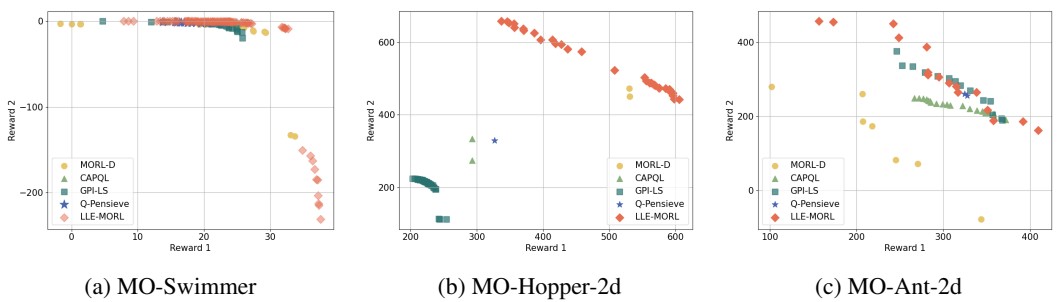

(a) MO-Swimmer  (b) MO-Hopper-2d  (c) MO-Ant-2d

Figure 5: Pareto fronts from the standard-training setting, comparing our LLE-MORL method with baselines on three continuous-control benchmarks. LLE-MORL consistently achieves wider coverage and closer proximity to the true Pareto front.

comprehensive understanding of algorithms' capabilities: **(1) Sample-Efficient Setting:** All methods, including our LLE-MORL approach, were trained for $1.5 \times 10^5$ timesteps. Given the complexity of continuous control benchmarks, this relatively limited interaction budget serves as a critical testbed for evaluating how rapidly different MORL strategies can discover effective Pareto front approximations. **(2) Standard-Training Setting:** To assess performance under more common training conditions for these continuous control benchmarks, most methods, including our LLE-MORL approach, were trained for $1 \times 10^6$ timesteps. An exception was made for the CAPQL baseline, which, due to its significant computational demands, was trained for $5 \times 10^5$ timesteps. This setting aligns with common practices for benchmarking in continuous control and allows us to assess the final quality of the Pareto fronts achieved by each algorithm after a more thorough learning process. Detailed training setups can be found in Appendix C.

First, we analyse performance in the sample-efficient setting, with results using Hypervolume (HV), Expected Utility (EU), and Sparsity (SP) metrics presented in Table 1 and the corresponding Pareto front visualisations in Figure 4. In this limited-interaction scenario, LLE-MORL achieves the highest HV and EU in all benchmarks, demonstrating strong capabilities in rapidly achieving high-quality Pareto fronts. Regarding SP, while LLE-MORL does not consistently achieve the leading scores on this metric, its performance generally reflects a good and effective distribution of solutions along the high-quality Pareto fronts it identifies. It should be noted that SP results can be confounded by a fragmentary recovery of the Pareto front. For instance, if Q-Pensieve discovers only two close points of the Pareto front for the MO-Ant problem, the sparsity rating is nearly perfect. Meanwhile, a low SP might also arise from solutions being overly clustered in a small region, as potentially seen with GPI-LS and Q-Pensieve in MO-Swimmer shown in Figure 4a.

Transitioning to the standard-training setting, where methods were trained for a more extensive duration, the evaluation results are presented in Table 2, and the corresponding Pareto front visualisation can be found in Figure 5. Across all benchmarks, LLE-MORL typically achieves the highest HV and highly competitive EU. This superior performance indicates that LLE-MORL finds a more extensive and higher-quality set of solutions, which strongly suggests a better approximation of the Pareto

| Environment | Metric | Method | | | | | |
|---|---|---|---|---|---|---|---|
| | | GPI-LS | CAPQL | Q-Pensieve | MORL/D | LLE-MORL-0 | LLE-MORL |
| MO-Swimmer | $HV(10^4)$ | 4.80±0.43 | 5.00±0.09 | 4.77±0.17 | 5.72±0.25 | 6.54±0.10 | **6.77±0.17** |
| | $EU(10^1)$ | 0.89±0.14 | 1.08±0.01 | 1.06±0.11 | 1.00±0.01 | 0.93±0.11 | **1.11±0.02** |
| | $SP(10^2)$ | 1.12±1.02 | 1.68±1.21 | **0.82±0.61** | 11.03±7.17 | 1.61±1.21 | 1.78±1.60 |
| MO-Hopper | $HV(10^5)$ | 1.06±0.12 | 1.40±0.31 | 1.00±0.05 | 1.96±0.19 | 2.73±0.17 | **3.02±0.28** |
| | $EU(10^2)$ | 2.11±0.17 | 2.68±0.40 | 2.16±0.08 | 3.29±0.15 | 4.15±0.20 | **4.36±0.28** |
| | $SP(10^2)$ | 5.35±3.64 | 1.33±1.21 | **0.08±0.04** | 48.25±22.90 | 8.08±6.44 | 15.13±8.31 |
| MO-Ant | $HV(10^4)$ | 3.24±1.28 | 8.10±0.64 | 5.01±1.73 | 9.73±0.42 | 10.07±0.63 | **10.44±0.56** |
| | $EU(10^2)$ | 0.72±0.30 | 1.82±0.10 | 1.18±0.41 | 2.03±0.08 | 2.16±0.10 | **2.23±0.08** |
| | $SP(10^3)$ | 1.14±1.01 | **0.05±0.03** | 0.12±0.10 | 1.33±1.02 | 1.51±0.36 | 1.22±0.22 |

Table 1: Sample-efficient evaluation of the quality of the Pareto front by hypervolume (HV), expected utility (EU) and sparsity (SP).

| Environment | Metric | Method | | | | | |
|---|---|---|---|---|---|---|---|
| | | GPI-LS | CAPQL | Q-Pensieve | MORL/D | LLE-MORL-0 | LLE-MORL |
| MO-Swimmer | $HV(10^4)$ | 5.46±0.17 | 5.12±0.32 | 5.20±1.08 | 7.05±0.39 | 7.74±0.22 | **7.82±0.24** |
| | $EU(10^1)$ | 1.06±0.03 | 1.10±0.05 | 1.13±0.15 | 1.13±0.03 | 1.03±0.07 | **1.16±0.09** |
| | $SP(10^2)$ | **0.04±0.02** | 0.12±0.09 | 0.05±0.02 | 4.36±1.64 | 1.11±0.56 | 1.15±0.63 |
| MO-Hopper | $HV(10^5)$ | 1.15±0.04 | 2.11±0.91 | 1.62±0.20 | 3.75±0.25 | 4.76±0.08 | **4.87±0.09** |
| | $EU(10^2)$ | 2.26±0.07 | 3.46±1.07 | 2.76±0.84 | 4.98±0.17 | 5.67±0.10 | **5.74±0.07** |
| | $SP(10^2)$ | 0.95±0.73 | 15.37±10.67 | **0.02±0.01** | 10.60±6.31 | 5.09±1.77 | 4.53±1.79 |
| MO-Ant | $HV(10^5)$ | 1.72±0.93 | 1.63±0.72 | 1.78±0.34 | 1.84±0.31 | 2.58±0.22 | **2.68±0.22** |
| | $EU(10^2)$ | 2.68±1.19 | 2.81±0.80 | 2.18±0.41 | 3.09±0.30 | 3.81±0.27 | **3.93±0.26** |
| | $SP(10^3)$ | 0.57±0.50 | 0.28±0.24 | **0.10±0.09** | 3.92±2.81 | 2.23±0.97 | 1.59±0.40 |

Table 2: Standard-training evaluation of the quality of Pareto front by hypervolume (HV), expected utility (EU) and sparsity (SP).

front compared to the baselines. The evidence from Pareto front visualisation further corroborates LLE-MORL's advantages. In the MO-Swimmer environment, shown in Figure 5a, LLE-MORL more comprehensively explores the objective space, successfully identifying Pareto optimal solutions in the lower-right region consistently missed by baselines such as GPI-LS, Q-Pensieve and CAPQL. Notably, when comparing the GPI-LS, CAPQL and Q-Pensieve performance to those in the sample-efficient setting for the MO-Swimmer environment, these particular baselines appear to remain constrained by suboptimal solutions in this challenging region, indicating that simply extending training duration did not resolve their exploration deficiencies here. While LLE-MORL's thorough exploration to achieve this broader coverage means its SP may not be the numerically lowest, the result could be well-justified by the extensive nature of the front.

In summary, LLE-MORL consistently demonstrates superior Pareto front approximations across both sample-efficient and standard-training evaluations. This robust performance is significantly supported by its innovative extension process, which is largely training-free once core parameter-performance relationships are established, allowing for the efficient generation of diverse and high-quality solutions. Consequently, LLE-MORL excels at both rapid learning in data-limited scenarios and achieving comprehensive, high-fidelity fronts with extended training, highlighting its distinct advantages for multi-objective reinforcement learning. Additionally, we present the running time of each algorithm in Appendix C, which shows the high efficiency level achievable by LLE-MORL.

### 4.3 ABLATION STUDY

The LLE-MORL integrates a locally linear extension process with a subsequent fine-tuning stage. To understand the distinct contributions of these components to the overall performance, our ablation study separates them. We first evaluate LLE-MORL-0, which solely employs the extension process without fine-tuning. As detailed in Table 1 and Table 2, LLE-MORL-0 itself demonstrates competitiveness, achieving strong Hypervolume (HV) and Expected Utility (EU) scores that are often competitive with or superior to baselines. This emphasises the efficacy of our extension mechanism in rapidly discovering a high-quality approximation of the Pareto front.

Subsequently, we assess the improvement of the fine-tuning stage by comparing LLE-MORL (which includes fine-tuning) to LLE-MORL-0. This comparison reveals that the inclusion of fine-tuning consistently yields further improvements in Hypervolume (HV) and Expected Utility (EU) across both sample-efficient and standard-training settings. The impact on Sparsity (SP) is less uniform, which is an expected outcome, as refining solutions towards a more optimal Pareto front can alter their relative spacing. Nevertheless, the consistent enhancements in HV and EU prove the value of

fine-tuning for improving the overall quality of the approximate Pareto front and its coverage by diverse solutions. This demonstrates that the extension process provides a strong foundation for the fine-tuning stage that enables LLE-MORL to outperform other algorithms.

## 5 RELATED WORK

Prior work in Multi-Objective Reinforcement Learning (MORL) offers various strategies for handling conflicting objectives. These can be broadly grouped into single-policy methods and multi-policy methods for approximating the Pareto front. Single-policy approaches, a foundational strategy in MORL, typically convert the multi-objective problem into a single-objective task using a predefined preference or weighting scheme to find a policy optimal for that specific trade-off. A common instance of such a weighting scheme is linear scalarization (Van Moffaert et al., 2013). Limitations of linear scalarization, particularly in capturing non-convex Pareto fronts, have been addressed by more advanced scalarization functions such as Chebyshev methods (Van Moffaert et al., 2013) and hypervolume-based approaches (Zhang & Golovin, 2020). Further theoretical work has aimed at enhancing scalarization robustness and performance, for instance, by proposing the addition of concave terms to rewards (Lu et al., 2023). Concurrently, significant efforts have developed generalised single-policy models conditioned on preference inputs to achieve adaptability across diverse objectives (Teh et al., 2017; Yang et al., 2019; Basaklar et al., 2022; Parisi et al., 2016), with subsequent extensions into offline learning contexts (Zhu et al., 2023; Lin et al., 2024) and methods to improve sample efficiency in these settings (Hung et al., 2023).

Multi-policy MORL strategies directly target the approximation of the entire Pareto front by learning a diverse collection of policies. One direction for generating diverse behaviours involves developing single, highly adaptable models conditioned on preferences, which generalise across various objectives using techniques like specialised experience replay or policy gradient methods that enforce Pareto stationarity (Abels et al., 2019; Friedman & Fontaine, 2018; Kyriakis & Deshmukh, 2022). Other approaches explicitly learn a diverse set of policies or their value functions; this includes direct value-based methods like Pareto Q-learning (Van Moffaert & Nowé, 2014), and evolutionary algorithms often guided by prediction models to discover a dense Pareto set (Xu et al., 2020). Further techniques for generating policy sets involve Generalised Policy Improvement (GPI) for sample-efficient learning (Alegre et al., 2023) or the development of transferable policy components using representations like successor features (Alegre et al., 2022). The use of constrained optimisation to efficiently complete and refine the Pareto front is also explored in (Liu et al., 2024; He et al., 2024). Furthermore, the principles of decomposition-based strategies, which find a set of solutions by solving multiple interrelated scalarised sub-problems, have been a significant focus, with recent work providing clarifying taxonomies and conceptual frameworks (Felten et al., 2024; Röpke et al., 2024).

While these established single-policy and multi-policy paradigms have significantly advanced MORL, the explicit characterisation and systematic exploitation of the structural relationship between the learned policies' underlying parameter space and their resultant performance on the Pareto front remain largely underexplored. Although multi-objective optimization offers techniques for navigating Pareto sets (Ye & Liu, 2022), and some MORL studies have touched upon parameter space regularities (Xu et al., 2020), policy manifolds (Parisi et al., 2016), or front geometries (Li et al., 2024), these explorations typically do not formalise or exploit the parameter-to-performance mapping for systematic, guided Pareto front generation.

## 6 CONCLUSION

We have discussed LLE-MORL, an algorithm that identifies solution components in multi-objective reinforcement learning by directed exploration of the Pareto front. The main benefit of LLE-MORL is increased efficiency which is enabled by maintaining a direct relation between the multi-objective performance and the representation of the policy in the parameter space. We have shown that this simple set-up is sufficient to obtain highly efficient coverage of a Pareto front as well as a better approximation of the Pareto front itself, so that we could show that LLE-MORL is superior to recent MORL algorithms. Furthermore, it can be expected that the approach can be extended to more than $d = 2$ objectives, although an implicit representation of the Pareto front may be preferable for $d > 2$ which would, however, reduce interpretability in terms of an accessible PPR as featured here.

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

# A ALGORITHM

In this section, we present a complete description of LLE-MORL, an efficient procedure for tracing an approximate Pareto front. Algorithm 1 details this process.

---

**Algorithm 1** LLE-MORL

---

**Require:** Initial scalarization weights $\{w_i\}_{i=1}^K$ evenly spanning preference space, Target scalarization weights for directional retraining $\{w_i'\}_{i=1}^K$, Initialization training length $T_{\text{init}}$, Directional retraining length $T_{\text{dir}}$, Fine-tuning length $T_{\text{ref}}$, Step-scale factors $\{\alpha_j\}_{j=1}^M$
**Ensure:** Approximate Pareto-optimal policy set $\Pi$

1: **Initialization:**
2: **for** $i = 1$ to $K$ **do**
3:      Train base policy $\theta_{w_i}$ with PPO under weight $w_i$ for $T_{\text{init}}$ steps
4: **end for**
5:
6: **Directional Retraining:**
7: **for** $i = 1$ to $K - 1$ **do**
8:      $\theta_{w_i'} \leftarrow$ continue training $\theta_{w_i}$ for $T_{\text{dir}}$ steps under $w_i'$
9:      $\Delta\theta_i \leftarrow \theta_{w_i'} - \theta_{w_i}$
10:     $\Delta w_i \leftarrow w_i' - w_i$
11: **end for**
12:
13: **Locally Linear Extension:**
14: $\mathcal{C} \leftarrow \emptyset$
15: **for** $i = 1$ to $K - 1$ **do**
16:      **for** $j = 1$ to $M$ **do**
17:          $\theta_{i,j} \leftarrow \theta_{w_i} + \alpha_j \Delta\theta_i$
18:          $w_{i,j} \leftarrow w_i + \alpha_j \Delta w_i$
19:          Evaluate performance $V(\theta_{i,j})$ under weight $w_{i,j}$
20:          $\mathcal{C} \leftarrow \mathcal{C} \cup \{(\theta_{i,j}, w_{i,j})\}$
21:      **end for**
22: **end for**
23:
24: **Candidate Selection:**
25: $\mathcal{N} \leftarrow$ non-dominated subset of $\mathcal{C}$
26:
27: **Preference-Aligned Fine-Tuning:**
28: $\mathcal{F} \leftarrow \emptyset$
29: **for all** $(\theta, w) \in \mathcal{N}$ **do**
30:      fine-tune $\theta$ for $T_{\text{ref}}$ steps under $w$, yielding $\theta'$
31:      add $(\theta', w)$ to $\mathcal{F}$
32: **end for**
33: $\mathcal{C}_{\text{all}} \leftarrow \mathcal{N} \cup \mathcal{F}$
34: $\mathcal{N}_{\text{final}} \leftarrow$ non-dominated subset of $\mathcal{C}_{\text{all}}$
35: $\Pi \leftarrow \Pi \cup \{\theta \mid (\theta, \cdot) \in \mathcal{N}_{\text{final}}\}$
36: **return** $\Pi$

---

The core pipeline involves several stages. First, $K$ base policies $\{\theta_{w_i}\}$ are trained, each under its respective initial weight $w_i$ for $T_{\text{init}}$ steps. Next, for the first $K - 1$ base policies, a short directional retraining is performed: each $\theta_{w_i}$ (for $i = 1 \ldots K - 1$) is further trained for $T_{\text{dir}}$ steps under its corresponding target weight $w_i'$ to yield $\theta_{w_i'}$. This allows the calculation of a parameter-space update vector $\Delta\theta_i = \theta_{w_i'} - \theta_{w_i}$ and the associated preference shift $\Delta w_i = w_i' - w_i$.

Using these $K - 1$ pairs of delta vectors, the Locally Linear Extension stage generates a set of candidate policies $\mathcal{C}$. For each original base policy $\theta_{w_i}$ (that had a corresponding $\Delta\theta_i$), intermediate candidates are formed by applying the scale factors $\alpha_j$ to $\Delta\theta_i$, also determining matched weights $w_{i,j}$. From this pool of generated candidates $\mathcal{C}$, a non-dominated subset $\mathcal{N}$ is selected. Policies in $\mathcal{N}$ then undergo Preference-Aligned Fine-Tuning for $T_{\text{ref}}$ steps under their matched weights, resulting

in a set of fine-tuned policies $\mathcal{F}$. Finally, the algorithm returns $\Pi$, which is the set of non-dominated policies selected from the combined pool of the initially selected non-dominated candidates $\mathcal{N}$ and their fine-tuned versions $\mathcal{F}$. The set $\Pi$ constitutes the approximated Pareto front.

# B  Theoretical Analysis

## B.1  Proof and Discussion of the Theorem

*Proof sketch.* By Assumption A1, the Pareto front is locally homeomorphic to Euclidean space, so tangent spaces exist. Assumption A3 ensures that for sufficiently small $\Delta\alpha$, local linear mappings between tangent spaces approximate the identity, meaning that extrapolation from nearby points remains close to the true manifold. Given $n = K$ distinct nearby solutions (Assumption A4, $K$ can be different from $n$ if a component of the Pareto front is dimension deficient), the tangent space at a base point can be spanned, allowing locally linear extension in all directions of the manifold. Repeated application of such steps covers the local neighborhood of the manifold, and resolution is controlled by the choice of $\Delta\alpha$.

The conditions for the applicability of the theorem do not generally preclude applicability of the algorithm, but clarify its limitations, i.e. cases where more effort and additional checks may become necessary:

L1: The Pareto front is not a (set of) manifold(s).

L2: The number of components of the Pareto front or its complexity exceeds any given bound.

L3: If the Pareto front is too complex, the algorithm may loose track. This can be mitigated by choosing a smaller $\Delta\alpha > 0$, although the efficiency of the algorithm will be reduced in this way.

L4: Less than $n$ different points of (a component of) the Pareto front are found to start with. If (the component of) the Pareto front is deficient in dimensionality, e.g. if (the component of) the Pareto front is discrete, then less than different $n$ points are sufficient for the algorithm.

## B.2  Proof and Discussion of the Proposition

In an $n$-objective reinforcement learning problem, if the Pareto front in the policy parameter space can be represented as a differentiable $(n-1)$-dimensional manifold, LLE-MORL can locally reconstruct this manifold by extending each base policy along $(n-1)$ distinct directions obtained via directional retraining. The locally linear extension then combines these directions to generate a grid of new candidate policies. In a $n$-objective task, $n-1$ directional retrained directions are used to form a $(n-1)$-dimensional grid: $\theta = \theta_{\text{base}} + \sum \alpha_i(\theta_i - \theta_{\text{base}})$, $i = 1, \ldots, n-1$, allowing dense coverage of the local Pareto surface with or without further training.

We restate Proposition from Section 3 for discussion convenience.

**Proposition.** *Time complexity. Given that the number of objectives is $n$, the number of base policies is $K$, the base policy training time is $T$, the number of locally linear extensions sampled per direction is $M$, and noting that the locally linear extension is training-free, the expected running time of LLE-MORL is $O(TK + KM^{n-1})$.*

*Proof.* Training $K$ base policies requires $O(TK)$. Around each base policy, local linear extension samples $M^{n-1}$ candidate policies across the $n-1$-dimensional tangent space. These extensions require no retraining, so their cost is dominated by evaluation, yielding the second term $O(KM^{n-1})$. Thus the overall runtime is $O(TK + KM^{n-1})$.

The first term in the time complexity proposition corresponds to the cost of training $K$ base policies, which is the dominant cost. The second term accounts for generating and evaluating $M^{n-1}$ extended policies per base policy. This step is training-free and its cost is controllable via $M = (\frac{\alpha_{\max} - \alpha_{\min}}{\Delta\alpha})$, allowing the user to balance Pareto front accuracy/density against computational resources. The second term grows exponentially with the number of objectives, but for low- to moderate-dimensional objectives, this cost remains small compared to policy training. For very high-dimensional objective spaces, adaptive sampling or sparse grids can mitigate this cost.

## C EXPERIMENT SETUP DETAILS

### C.1 BENCHMARKS

To evaluate the performance of our proposed LLE-MORL method and compare it against existing baselines, we utilise a suite of continuous control benchmarks from the MO-Gymnasium library (Felten et al., 2023). These environments are designed to test the ability of an agent to learn policies that effectively balance multiple, often conflicting objectives. The specific environments and their multi-objective reward formulations are detailed below:

**MO-SWIMMER-V5.** A planar, three-link swimmer operating in a viscous fluid, utilising a 2D continuous action space to control its joint torques. The objectives are to maximise forward velocity along the $x$-axis and minimise the control cost.

The observation space $\mathcal{S} \subset \mathbb{R}^8$ includes joint angles and velocities, and the action space $\mathcal{A} \subset \mathbb{R}^2$ represents joint torques in $[-1, 1]$. Let $x_{\text{before}}$ and $x_{\text{after}}$ be the $x$-coordinates of the centre of mass of swimmer before and after an action, $\Delta t$ be the time step, and $a_j$ be the $j$-th component of the action vector.

The first objective is the forward speed

$$R_1 = \frac{x_{\text{after}} - x_{\text{before}}}{\Delta t},$$

and the second objective is the energy efficiency (negative control cost):

$$R_2 = -\sum_j a_j^2$$

**MO-HOPPER-2OBJ-V5.** This environment features a 2D one-legged hopper with a 3-dimensional continuous action space controlling torques for its thigh, leg, and foot joints. Originally a 3-objective task (forward speed, jump height, control cost), we use the 2-objective variant, in which the separate control-cost objective is added to other objectives.

The observation space $\mathcal{S} \subset \mathbb{R}^{11}$ includes joint states and torso position, and the action space $\mathcal{A} \subset \mathbb{R}^3$ represents joint torques in $[-1, 1]$. Let $v_x = (x_{\text{after}} - x_{\text{before}})/\Delta t$ be the forward velocity of the agent along the $x$-axis, where $x_{\text{after}}$ and $x_{\text{before}}$ are x-positions of the torso. Let $h_{\text{jump}} = 10 \times (z_{\text{after}} - z_{\text{init}})$ be a measure of jumping height, where $z_{\text{after}}$ is the current z-position of the torso and $z_{\text{init}}$ is its initial z-position. Let $c_{\text{ctrl}}$ be the positive control cost, computed as $w_{\text{env\_ctrl}} \sum_j (a_j)^2$, where $w_{\text{env\_ctrl}}$ is the environment control cost weight (typically 0.001). Let $r_{\text{healthy}}$ be the health reward (typically $+1$ if the agent has not fallen). The reward vector $\mathbf{R} = [R_1, R_2]$ is defined as:

- $R_1$ (Adjusted Forward Performance):
$$R_1 = v_x + r_{\text{healthy}} - c_{\text{ctrl}}$$

- $R_2$ (Adjusted Height Performance):
$$R_2 = h_{\text{jump}} + r_{\text{healthy}} - c_{\text{ctrl}}$$

**MO-ANT-2OBJ-V5.** A quadrupedal "ant" robot in 2D with an eight-dimensional action space for joint torques. By default, the environment emits a three-dimensional reward vector: (1) $x$-velocity, (2) $y$-velocity, and (3) control cost. Here, we use the two-objective variant in which the separate control-cost objective is added to other objectives.

The observation space $\mathcal{S} \subset \mathbb{R}^{27}$ includes joint states, torso position, and contact forces, and the action space $\mathcal{A} \subset \mathbb{R}^8$ represents joint torques in $[-1, 1]$. Let $v_x = (x_{\text{after}} - x_{\text{before}})/\Delta t$ be the forward velocity of the agent along the $x$-axis, where $x_{\text{after}}$ and $x_{\text{before}}$ are x-positions of the torso. Let $v_y = (y_{\text{after}} - y_{\text{before}})/\Delta t$ be the forward velocity of the agent along the $y$-axis, where $y_{\text{after}}$ and $y_{\text{before}}$ are y-positions of the torso. Let $c_{\text{ctrl}}$ be the positive control cost, computed as $w_{\text{env\_ctrl}} \sum_j (a_j)^2$, where $w_{\text{env\_ctrl}}$ is the environment control cost weight (typically 0.05). Let $r_{\text{healthy}}$ be the health reward (typically $+1$ if the Ant is healthy). Let $p_{\text{contact}}$ be the positive contact penalty, which is used for penalising the Ant if the external contact forces are too large, computed as $w_{\text{env\_contact}} \sum_k (\text{force}_k)^2$, where $w_{\text{env\_contact}}$ is the environment contact cost weight (typically $5 \times 10^{-4}$). The reward vector $\mathbf{R} = [R_1, R_2]$ is defined as:

- $R_1$ (Adjusted $x$-Velocity Performance):

$$R_1 = v_x + r_{\text{healthy}} - c_{\text{ctrl}} - p_{\text{contact}}$$

- $R_2$ (Adjusted $y$-Velocity Performance):

$$R_2 = v_y + r_{\text{healthy}} - c_{\text{ctrl}} - p_{\text{contact}}$$

## C.2 EVALUATION METRICS

We evaluate the quality of the approximate Pareto front using three standard metrics, following the formalism in (Zitzler & Thiele, 2002; Zintgraf et al., 2015; Hayes et al., 2022).

**Hypervolume (HV).** Let $P$ be an approximate Pareto front and $G_0$ a reference point dominated by all $p \in P$. The hypervolume is $\mathcal{H}(P) = \int_{\mathbb{R}^d} \mathbb{1}_{H(P)}(z)\, dz$, where $H(P) = \{\, z \in \mathbb{R}^n \mid \exists j,\ 1 \leq j \leq |P| \ :\ G_0 \preceq z \preceq P(j)\}$. Here, $P(j)$ is the $j^{\text{th}}$ solution in $P$, the symbol $\preceq$ denotes objective dominance, and $\mathbb{1}_{H(P)}$ is an indicator function that equals 1 if $z \in H(P)$ and 0 otherwise. A higher hypervolume implies a front closer to and more extensive with respect to the true Pareto front.

**Expected Utility (EU).** Let $P$ be an approximate Pareto front and $\Pi$ be the corresponding policy set. The expected utility metric is $\mathcal{U}(P) = \mathbb{E}_{\omega \sim \Omega}\left[\max_{\pi \in \Pi}\ \omega^\top G_\omega^\pi\right]$. A higher EU denotes better average performance over preferences.

**Sparsity (SP).** Let $P$ be an approximate Pareto front in a $d$-dimensional objective space. The sparsity metric is $S(P) = \frac{1}{|P|-1} \sum_{i=1}^{d} \sum_{k=1}^{|P|-1} \left(\tilde{G}_i(k) - \tilde{G}_i(k+1)\right)^2$, where $\tilde{G}_i$ is the sorted list of the $i^{\text{th}}$ objective values in $P$, and $\tilde{G}_i(k)$ is the $k^{\text{th}}$ entry in this sorted list. Lower sparsity indicates a more uniform distribution of solutions along each objective.

## C.3 HUNGARIAN MATCHING DISTANCE

To measure structural similarity between policies, we use the Hungarian matching distance (Kuhn, 1955; Munkres, 1957).

**Definition 4.** *Hungarian matching distance. For a given layer $l$ with neuron sets $A^{(l)}$ and $B^{(l)}$, let $w_i^{(l)}$ and $w_j^{(l)}$ denote the incoming weight vectors of neurons $i \in A^{(l)}$ and $j \in B^{(l)}$, respectively. The minimum-cost perfect matching $M^{(l)}$ between $A^{(l)}$ and $B^{(l)}$ is obtained by using the Hungarian algorithm:*

$$M^{(l)} = \underset{matching}{\arg\min} \sum_{(i,j) \in matching} \left\| w_i^{(l)} - w_j^{(l)} \right\|_2 \tag{1}$$

*The Hungarian matching distance between the two networks is then defined as*

$$d_{Hungarian}(A, B) = \sum_{l=1}^{L} \sum_{(i,j) \in M^{(l)}} \left\| w_i^{(l)} - w_j^{(l)} \right\|_2 \tag{2}$$

*where $L$ is the total number of layers.*

## C.4 SANITY CHECK DETAILS

In Section 3.3, the sanity check was designed as a qualitative validation of the parameter–performance relationship (PPR), rather than a full experiment.

To construct policy pairs, we trained 11 base policies $\theta_{\omega 1}$ using the scalarization vector $\omega_1$ that were equally distributed in the preference space, i.e. $\omega_1 = [1, 0], [0.9, 0.1], ..., [0, 1]$.

For each base policy, we formed a paired scalarization vector $\omega_2$ by shifting $\omega_1$ with a retraining shift parameter $\delta_s$:

$$\omega_2 = \begin{cases} [\omega_{11} - \delta_s, \omega_{12} + \delta_s], & \text{if } \omega_{11} - \delta_s \in [0, 1] \\[2mm] [\omega_{11} + \delta_s, \omega_{12} - \delta_s], & \text{otherwise} \end{cases} \tag{3}$$

where $\delta_s \in \{0.1, 0.2, 0.3, 0.4, 0.5\}$. This procedure yielded multiple pairs of $(\omega_1, \omega_2)$ across the preference space. For each pair, we performed short retraining from $\theta_{\omega_1}$ under $\omega_2$ and compared the orignial $\theta_{\omega_1}$ and resulting model $\theta_{\omega'}$ with the independently trained $\theta_{\omega_2}$. This allowed us to consistently observe that retrained policies stay close in parameter space (low Hungarian distance), while having directional movement in performance space, supporting the local PPR hypothesis. Figure 2 in the main paper presents a representative example from these trials.

## C.5 TRAINING DETAILS

All learning phases within our LLE-MORL algorithm, including the initial training of base policies, the directional retraining, and the final preference-aligned fine-tuning, utilize the Proximal Policy Optimization (PPO) algorithm (Schulman et al., 2017). We employed a standard PPO implementation from the Stable Baselines3 library (Raffin et al., 2019). The PPO parameters used across all training stages and benchmarks are detailed in Table 3.

The specific parameters for the LLE-MORL pipeline include:

- **Number of Initial Base Policies** ($K$): The total count of base policies $\theta_{w_j}$, trained in the initialization stage. The corresponding $K$ initial scalarization weights $\{w_i\}_{i=1}^{K}$ are generated by evenly distributing them across the preference space (e.g., for 2D objectives, from $[1, 0]$ to $[0, 1]$ in $K$ steps).

- **Initialization Training Timesteps** ($T_{\text{init}}$): The number of environment interaction steps for which the initial base policy $\theta_{w_i}$ is trained under its weight $w_i$.

- **Retraining Preference Shift Strategy (controlled by shift magnitude $\delta_s$)**: Target scalarization weights $\{w_i'\}$ for directional retraining are generated by shifting each initial weight $w_i$ to a nearby, distinct point on the preference space. The extent of this shift is controlled by a hyperparameter $\delta_s$. Conceptually, for $d$-dimensional preference spaces ($d > 2$), this shift could be defined as an angular displacement in the space. In our current two-dimensional objective experiments ($d = 2$), where $w_i = [w_{i,0}, w_{i,1}]$, this shift is implemented by moving the first component $w_{i,0}$ by the magnitude $\delta_s$ to obtain $w_{i,0}'$. The shift direction (decrease or increase) is chosen to keep the component within valid bounds (e.g., $[0, 1]$), and the default direction is decrease; then the second component $w_{i,1}$ is adjusted accordingly (assuming all objective weights sum to 1).

- **Directional Retraining Timesteps** ($T_{\text{dir}}$): The number of environment interaction steps for which the base policy $\theta_{w_i}$ is retrained under its target weight $w_i'$ to produce $\theta_{w_i'}$.

- **Step-Scale Factor Generation** ($\alpha_{\text{start}}, \alpha_{\text{end}}, \Delta\alpha$): The set of step-scale factors $\{\alpha_j\}$ used in Locally Linear Extension is generated based on a starting value ($\alpha_{\text{start}}$), an ending value ($\alpha_{\text{end}}$), and either a step increment ($\Delta\alpha$).

- **Fine-tuning Timesteps** ($T_{\text{ref}}$): The number of environment interaction steps for which the selected candidate policy from the extension phase is fine-tuned under its matched preference weight $w_{i,j}$.

The specific values for these LLE-MORL parameters, are provided in Table 4 and Table 5.

| Parameter Name | MO-Swimmer | MO-Hopper-2d | MO-Ant-2d |
|---|---|---|---|
| steps per actor batch | 512 | 512 | 512 |
| learning rate ($\times 10^{-4}$) | 3 | 3 | 3 |
| learning rate decay ratio | 1 | 1 | 1 |
| $\gamma$ | 0.995 | 0.995 | 0.995 |
| GAE lambda | 0.95 | 0.95 | 0.95 |
| number of mini batches | 32 | 32 | 32 |
| PPO epochs | 10 | 10 | 10 |
| entropy coefficient | 0.0 | 0.0 | 0.0 |
| value loss coefficient | 0.5 | 0.5 | 0.5 |
| maximum gradient norm | 0.5 | 0.5 | 0.5 |
| clip parameter | 0.2 | 0.2 | 0.2 |

Table 3: PPO hyperparameters for benchmarks.

| Parameter Name | Symbol | MO-Swimmer | MO-Hopper-2d | MO-Ant-2d |
|---|---|---|---|---|
| Number of base policies | $K$ | 6 | 6 | 6 |
| Initialization timesteps | $T_{\mathrm{init}}$ | $1 \times 10^5$ | $1 \times 10^5$ | $1 \times 10^5$ |
| Preference shift magnitude | $\delta_s$ | 0.1 | 0.1 | 0.1 |
| Directional retraining timesteps | $T_{\mathrm{dir}}$ | $1 \times 10^4$ | $1 \times 10^4$ | $1 \times 10^4$ |
| Step-scale start | $\alpha_{\mathrm{start}}$ | -1.5 | -1.5 | -1.5 |
| Step-scale end | $\alpha_{\mathrm{end}}$ | 1.5 | 1.5 | 1.5 |
| Step-scale increment | $\Delta\alpha$ | 0.05 | 0.05 | 0.05 |
| Fine-tuning timesteps | $T_{\mathrm{ref}}$ | $1 \times 10^3$ | $1 \times 10^3$ | $1 \times 10^3$ |

Table 4: Hyperparameters for the LLE-MORL across benchmarks under sample-efficient setting.

| Parameter Name | Symbol | MO-Swimmer | MO-Hopper-2d | MO-Ant-2d |
|---|---|---|---|---|
| Number of base policies | $K$ | 6 | 6 | 6 |
| Initialization timesteps | $T_{\mathrm{init}}$ | $1 \times 10^6$ | $1 \times 10^6$ | $1 \times 10^6$ |
| Preference shift magnitude | $\delta_s$ | 0.1 | 0.1 | 0.1 |
| Directional retraining timesteps | $T_{\mathrm{dir}}$ | $1 \times 10^4$ | $1 \times 10^4$ | $1 \times 10^4$ |
| Step-scale start | $\alpha_{\mathrm{start}}$ | -1.5 | -1.5 | -1.5 |
| Step-scale end | $\alpha_{\mathrm{end}}$ | 1.5 | 1.5 | 1.5 |
| Step-scale increment | $\Delta\alpha$ | 0.05 | 0.05 | 0.05 |
| Fine-tuning timesteps | $T_{\mathrm{ref}}$ | $1 \times 10^3$ | $1 \times 10^3$ | $1 \times 10^3$ |

Table 5: Hyperparameters for the LLE-MORL across benchmarks under standard-training setting.

### C.6 COMPUTATIONAL RESOURCES

All experiments were run on a workstation equipped with an AMD Ryzen Threadripper PRO 5975WX (32 cores), an NVIDIA GeForce RTX 3090 GPU (24 GB GDDR6X), and 256 GiB of RAM, running Ubuntu 24.04 LTS. The software stack included CUDA Toolkit 12.0 and the corresponding NVIDIA drivers. Approximate execution times for all methods and benchmarks are reported separately in Table 6.

## D STATISTICAL SIGNIFICANCE OF THE EXPERIMENTAL RESULTS

In the LLE-MORL algorithmic procedure, the underlying Reinforcement Learning method (PPO) used in the training and retraining phases is subject to inherent stochasticity arising from factors such as random seed initialisation for neural network weights and data sampling during training. This means that while the overarching behaviour and effectiveness of LLE-MORL are generally reproducible, the exact set of policies discovered on the approximated Pareto front, their specific parameter values, or their ordering might exhibit some variation across independent runs started with different random seeds. Consequently, direct averaging of entire Pareto fronts or performing straightforward statistical tests on the precise composition of these policy sets can be non-trivial and may not always be the most informative way to capture the consistent ability of the algorithm to find high-quality solution regions.

Our demonstration of reliability and robustness relies on: (1) the consistent observation of the superior performance of LLE-MORL in generating high-quality Pareto fronts compared to baselines; (2) sensitivity analyses of key hyperparameters (detailed in Appendix F), which show consistent outcome patterns within certain parameter ranges; (3) the qualitative consistency in the shape, extent, and dominance characteristics of the visualized Pareto fronts (e.g. Figure 4 and Figure 5); and (4) the quantitative results reported in Table 4 and Table 5 across multiple seeds, further reinforcing the statistical reliability of our findings. These combined observations support the robustness of our conclusions and the effectiveness of LLE-MORL.

## E LIMITATIONS

Limitations of our approach are implied by inherent challenges in multi-objective optimisation, but we also note some limitations that are specific to our algorithm and require further study.

| Method | Sample-efficient setting | | | Standard-training setting | | |
|---|---|---|---|---|---|---|
| | MO-Swimmer | MO-Hopper-2d | MO-Ant-2d | MO-Swimmer | MO-Hopper-2d | MO-Ant-2d |
| GPI-LS | 3 | 3 | 3 | 22 | 23 | 19 |
| CAPQL | 13 | 8 | 12 | 95 | 77 | 68 |
| Q-Pensieve | 3 | 3 | 3 | 17 | 15 | 18 |
| MORL/D | 1 | 1 | 1 | 3 | 3 | 3 |
| LLE-MORL | 1 | 1 | 1 | 3 | 3 | 3 |

Table 6: Approximate execution times (hours) for each method and benchmark under sample-efficient and standard training setting.

- We have restricted ourselves to problems with two objectives where the Pareto front is one-dimensional. A larger number of objectives is a problem for most of the existing MORL algorithms. Although usually some of the objectives are of different importance and can be lexicographically ranked, so that the complexity does not necessarily increase exponentially with the number of objectives. The high-dimensional case is nevertheless challenging, but our approach can be seen as promising: In higher dimensions, the number of solutions that are in the same local quasi-linear patch increases dramatically, so that the efficiency of the proposed local search will be even more beneficial. This benefit could be reduced by the potentially increasing complexity of the topological relation between the performance space and the parameter space which could be a fascinating subject for future work.

- We are assuming that the Pareto front consists of a relatively small number of connectivity components which have a manifold structure. While there is no theoretical bound to the complexity of the Pareto from, the idea of MORL implies that the objectives are at least in some sense comparable. For Pareto fronts that are fractal or of high genus, the result of multi-objective optimisation lacks robustness, although it will neither be possible to fix any limits for the complexity of the Pareto front. However, as long as there are only a limited number of manifold-like connectivity components, our algorithm will be applicable.

- Widely different scales and elasticities of the objectives can lead to problems as in optimisation in anisotropic error landscapes. Step size control that helps in gradient methods in optimisation, will also be useful here, but has not been studied yet, as the typical (benchmark) problems are sufficiently isotropic.

- The density of the identified solutions on the Pareto front is clearly a challenge which may be solved by step size control as mentioned in the previous point. This concerns higher-dimensional cases as well as extended one-dimensional trails as visible in the top trail in Figure 3c also a simple reduction of the parameter $\Delta\alpha$ at the observation of large steps in the performance space could have solved this issue already so that a more uniform covering of the Pareto from is not difficult to achieve in the present approach. See also Appendix F.2. In contrast to other approaches, linear regions of the Pareto can trivially be tracked by LLE-MORL. Concave regions connected to the Pareto front will be followed through without problem, but will need to be removed in a single postprocessing step as they are dominated by other solutions. Even full patches of solutions may turn out to be Pareto sub-optimal and require a similar treatment.

- We have made use of scalarisation to seed the solution domains, whereas the reconstruction of the Pareto front is done by a lateral process that does not depend on preference weights. It is in principle possible that a solution patch is not reachable by any scalarisation-based seeding attempt, see also the early discussion in (Vamplew et al., 2008). In this case our approach might not find this patch, although it is still possible that it is found by retraining from a different solution domain as shown in Figure 3c.

# F  ADDITIONAL RESULTS

## F.1  EFFECT OF DIRECTIONAL RETRAINING SHIFT

This ablation study investigates the influence of the directional retraining shift $\delta_s$, on the Locally Linear Extension (LLE) process. We analyse the performance of the LLE-MORL-0 variant (which excludes fine-tuning) under the standard-training setting. We vary $\delta_s$ over the set $\{0.1, 0.2, 0.3, 0.4, 0.5\}$,

| Environment | Metric | $\delta_s = 0.1$ | $\delta_s = 0.2$ | $\delta_s = 0.3$ | $\delta_s = 0.4$ | $\delta_s = 0.5$ |
|---|---|---|---|---|---|---|
| MO-Swimmer | $HV(10^4)$ | 7.37 | 7.38 | 7.38 | 7.36 | 7.37 |
| | $EU(10^1)$ | 1.03 | 1.29 | 1.41 | 0.99 | 1.60 |
| | $SP(10^2)$ | 2.10 | 0.83 | 3.05 | 2.39 | 2.12 |
| MO-Hopper-2d | $HV(10^5)$ | 4.88 | 4.84 | 4.79 | 4.86 | 4.80 |
| | $EU(10^2)$ | 5.65 | 5.56 | 5.49 | 5.54 | 5.58 |
| | $SP(10^2)$ | 7.20 | 4.49 | 3.33 | 2.54 | 3.33 |
| MO-Ant-2d | $HV(10^5)$ | 2.43 | 2.66 | 2.65 | 2.72 | 2.44 |
| | $EU(10^2)$ | 3.44 | 3.94 | 3.69 | 3.95 | 3.66 |
| | $SP(10^3)$ | 1.76 | 2.64 | 2.23 | 8.98 | 2.41 |

Table 7: Impact of retraining shift $\delta_s$ on LLE-MORL-0 performance (HV, EU, SP) under standard-training setting.

| Environment | Metric | $(\alpha_{start}, \alpha_{end})$ $\Delta\alpha$ | (-1,1) 0.01 | 0.05 | 0.1 | 0.5 | (-1.5,1.5) 0.01 | 0.05 | 0.1 | 0.5 | (-2,2) 0.01 | 0.05 | 0.1 | 0.5 | (-3,3) 0.01 | 0.05 | 0.1 | 0.5 |
|---|---|---|---|---|---|---|---|---|---|---|---|---|---|---|---|---|---|---|
| MO-Swimmer | $HV(10^4)$ | | 7.37 | 7.37 | 7.37 | 7.33 | 7.38 | 7.37 | 7.37 | 7.33 | 7.40 | 7.40 | 7.39 | 7.34 | 7.40 | 7.40 | 7.40 | 7.34 |
| | $EU(10^1)$ | | 1.08 | 1.27 | 1.32 | 1.33 | 1.08 | 1.03 | 1.42 | 1.65 | 1.13 | 1.41 | 1.53 | 1.70 | 1.43 | 1.75 | 1.73 | 2.37 |
| | $SP(10^2)$ | | 0.24 | 1.17 | 2.54 | 25.55 | 1.13 | 2.10 | 1.46 | 25.55 | 0.11 | 0.53 | 1.18 | 11.48 | 0.11 | 0.51 | 1.18 | 11.48 |
| MO-Hopper-2d | $HV(10^5)$ | | 4.95 | 4.88 | 4.85 | 4.80 | 4.96 | 4.88 | 4.90 | 4.80 | 4.95 | 4.88 | 4.88 | 4.81 | 4.96 | 4.88 | 4.88 | 4.81 |
| | $EU(10^2)$ | | 5.82 | 5.74 | 5.73 | 5.75 | 5.76 | 5.65 | 5.72 | 5.75 | 5.75 | 5.69 | 5.69 | 5.67 | 5.75 | 5.69 | 5.69 | 5.68 |
| | $SP(10^2)$ | | 5.63 | 9.43 | 11.58 | 60.35 | 5.01 | 7.20 | 13.06 | 60.35 | 3.02 | 5.42 | 10.05 | 29.66 | 3.62 | 5.41 | 10.05 | 29.66 |
| MO-Ant-2d | $HV(10^5)$ | | 2.57 | 2.43 | 2.40 | 2.23 | 2.60 | 2.43 | 2.23 | 2.14 | 2.60 | 2.47 | 2.25 | 2.14 | 2.60 | 2.47 | 2.25 | 2.14 |
| | $EU(10^2)$ | | 3.61 | 3.47 | 3.62 | 3.66 | 3.66 | 3.44 | 3.44 | 3.31 | 3.67 | 3.59 | 3.43 | 3.37 | 3.67 | 3.59 | 3.43 | 3.37 |
| | $SP(10^3)$ | | 0.54 | 2.03 | 3.11 | 7.92 | 1.85 | 1.76 | 1.79 | 8.36 | 1.85 | 1.55 | 1.79 | 8.36 | 1.85 | 1.55 | 1.79 | 8.36 |

Table 8: Influence of Step-Scale $\alpha$ on LLE-MORL-0 performance under standard-training setting.

keeping other parameters at their default values, and report Hypervolume (HV), Expected Utility (EU), and Sparsity (SP) for each environment in Table 7.

The results indicate that $\delta_s$ influences the resulting Pareto front approximation, as different shift values generate distinct extension trajectories that directly shape the front. While optimal performance varies across environments, a moderate retraining shift (e.g., $\delta_s$ in the range of 0.2 to 0.3, depending on the specific benchmark characteristics evident in Table 7) generally appears to strike an effective balance between front coverage (HV, EU) and solution diversity (SP). Our default configuration utilized $\delta_s = 0.1$, chosen for its simplicity and promising results in preliminary tests. However, this ablation demonstrates that this typical shift is not universally optimal; selecting an appropriately tuned moderate $\delta_s$ can further enhance the quality and coverage of the extended manifold, thereby improving the final Pareto front approximation achieved by LLE-MORL-0.

## F.2 EFFECT OF STEP-SCALE IN LOCALLY LINEAR EXTENSION

We next examine how the choice of step-scale factors $\{\alpha_j\}$ — defined by their start $\alpha_{start}$, end $\alpha_{end}$, and increment $\Delta\alpha$ — influences the performance of LLE-MORL-0. For each benchmark, we compare different ranges and resolutions of $\{\alpha_j\}$ and report HV, EU, and SP in Table 8.

The choice of $\Delta\alpha$ significantly impacts Sparsity (SP), as shown in Table 8. Finer increments (smaller $\Delta\alpha$) lead to substantially lower SP values, indicating denser Pareto front approximations, whereas coarser steps result in sparser solutions. This confirms that LLE offers a training-free mechanism to control the density of the approximated front simply by adjusting $\Delta\alpha$.

Regarding the range of $\alpha_j$ (controlled by $\alpha_{start}$ and $\alpha_{end}$), expanding it generally leads to improvements in both HV and EU, as the extension manifold can reach further from the initial policy along the identified directional vector (visualized conceptually in Figure 3). However, Table 8 suggests that an extremely wide range does not always yield proportional gains in coverage. This indicates that beyond a certain point, the linearity assumption underpinning LLE may become less effective for extending the front into novel, high-quality regions using a single directional vector, or that the most valuable regions reachable by the current $\Delta\theta$ vectors are already sufficiently captured.

# G    USE OF LARGE LANGUAGE MODELS

In accordance with the ICLR 2026 policy on the use of Large Language Models (LLMs), we claim that ChatGPT (OpenAI) and Gemini (Google DeepMind) were used to assist with only writing polishing. The scientific content, algorithms, experiments, and analysis were fully conceived, implemented, and validated by the authors.

