# OpenReview forum: "LLE-MORL: Locally Linear Extrapolation of Policies for Efficient Multi-Objective Reinforcement Learning"
_ICLR.cc/2026/Conference — ICLR 2026 Conference Withdrawn Submission_

### Official Review · Reviewer_Qph7 · 2025-10-29

**Soundness:** 1
**Presentation:** 2
**Contribution:** 2
**Rating:** 0
**Confidence:** 5

**Summary:**

The paper proposes a "parameter-performance relationship" (PPE) in multi-objective reinforcement learning (MORL). PPE is an empirical insight that small shifts in the parameter space result in predictable small shifts in the performance space. The paper formalizes this insight and devises an algorithm, called LLE-MORL, that exploits it for more sample-efficient Pareto front exploration in MORL. The algorithm works by first searching an initial set of policies using PPO. Then, these policies are retrained to get directions for extrapolation. Further points on the Pareto front are then obtained using linear inter- and extrapolation of the weight vectors from the initial to retrained ones. A final fine-tuning stage then ensures that the extrapolated policies stay close to the Pareto front.

The paper poses a theorem that LLE-MORL will faithfully reconstruct the Pareto front up to a certain precision if some assumptions are met. LLE-MORL is then empirically evaluated against multiple baselines on standard MORL environments and found to be consistently superior.

**Strengths:**

- The idea that parameter space changes are in a structured way related to changes in the space of returns is a fertile one
- The proposed algorithm enjoys high sample-efficiency due to interpolating the policies on the Pareto front rather than searching for them from scratch
- A comprehensive set of environments and tracked metrics is selected when evaluating the method1

**Weaknesses:**

# Theory
The theoretical discussion in the paper is not sound.

First, the Parameter-Performance Relationship (PPR) is defined in L182-186 in a way that makes it clear it will almost never hold in practice. The key problem is that the value $h(\Delta\theta)$ in the definition does not depend on $\theta$, only on $\Delta \theta$. Even if $V(\theta+\Delta\theta)-V(\theta)=h(\Delta\theta)$ only holds when $\Vert\Delta\theta\Vert<\delta$, this implies that the function $V(\theta)$ is _globally affine_. To see that,
$$h(\Delta\theta_1+\Delta\theta_2)=V(\theta+\Delta\theta_1+\Delta\theta_2)-V(\theta)=[V(\theta+\Delta\theta_1+\Delta\theta_2)-V(\theta+\Delta\theta_1)]+[V(\theta+\Delta\theta_1)-V(\theta)]=h(\Delta\theta_2)+h(\Delta\theta_1).$$
So $h$, at least in a small ball, satisfies the Cauchy functional equation. Then, under mild regularity conditions, it is linear. We can then tile the entire parameter space with overlapping balls of radius $<\delta$ to show global linearity. This implies that $V
(\theta)=V(\mathbf{0})+h(\theta)$ is affine. This in turn implies that straight lines in the parameter space should translate into straight lines in the performance space, which is clearly not true even from Fig. 3 in the paper (the curves on the figure are not straight). Such affinity would also break under reparameterization. Even in the simplest case where a tabular policy is parameterized directly by action probabilities, the value function depends non-linearly on the policy, see for example Fig. 4 in [1].

What the authors could potentially have in mind is that the function $V(\theta)$ is often smooth, but this would be a much weaker statement that would still require additional scrutiny.

Second, the theorem on L286-289 does not stand to the standards of mathematical rigor. The assumptions A1-A4 are only defined in sketches. It is not stated what it would mean for the algorithm to reconstruct a part of the Pareto front "to resolution $\Delta\alpha$." Furthermore, even if we ignore this omission, we can always set $\Delta\alpha$ to be large enough for the theorem to be trivially satisfied. The assumptions 1-4 are also only stated in natural language, without proper mathematical notation. They also seem to contradict each other. In particular, a differentiable manifold is standardly defined through differentiable (hence continuous) maps from Euclidean space, meaning that it would of course contain what the paper refers to as "cluster points" beyond some edge cases.  The "proof," deferred to the appendix, turns out to be a three-paragraph proof sketch.

[1] Dadashi, Robert, et al. "The value function polytope in reinforcement learning." International Conference on Machine Learning. PMLR, 2019.

# Missed prior work
The idea behind LLE-MORL that we can linearly interpolate in the reward space to get the nearly-Pareto optimal rewards has also been proposed in the literature under the name of "reward soups" [2]. The paper regrettably fails to cite this prior work. As it stands, LLE-MORL could be seen as an incremental improvement to the method of reward soups evaluated on different environments ([2] works with text and image generation), but the novelty is highly limited.

[2] Rame, Alexandre, et al. "Rewarded soups: towards pareto-optimal alignment by interpolating weights fine-tuned on diverse rewards." Advances in Neural Information Processing Systems 36 (2023): 71095-71134.

# Problems with experimental evaluation
RL algorithms in general, and especially MORL, are known to be particularly sensitive to hyperparameters. From the experimental sections in the paper and in the appendix, I cannot see any attempts to tune the hyperparameters of the methods. I understand that the computational resources are limited (from table 6), but in such cases it is standard to do hyperparameter tuning in a reduced setup, e.g. when running for fewer total timesteps. I couldn't find the hyperparameters that were selected for any of the baselines either.

I am especially suspicious that the paper fails to tune the hyperparameters correctly in light of the results in Figure 3. It claims to obtain the "original policy" weights by running PPO "to convergence." However, Figure 3 (c) reveals that the original policy is highly Pareto-non optimal (regardless of relative reward weights), since there is a better policy on the top right of the figure.

In this form, the paper cannot be accepted to the conference.

**Questions:**

- What are the x- and y-axes of the heatmap in Figure 2 (a)?
- When defining Hungarian distance, how do you efficiently search for permutations that make the networks closest?

---

> ### Author Response · Authors · 2025-12-03
>
> Thanks for the opportunity to provide some clarification for our work.
>
> ## Prior work ##
> While rewarded soups and LLE-MORL both manipulate policy parameters, our approach provides a fundamentally different capability that those weighted aggregation methods cannot offer, the ability to locally capture and efficiently explore the Pareto front manifold.
>
> Rewarded soups is a decomposition-based method which utilises weighted aggregation approaches between multiple trained single-objective policies. This can combine discrete solutions but cannot directly explore or reconstruct the local Pareto front manifold between these solutions.
> This limitation arises because each policy is independently optimized for a different single-objective, it naturally lacks information about how small changes in policy parameters affect performance across multiple objectives, which is crucial for accurately tracing the local manifold of the Pareto front.
>
> Moreover, the mapping from policy parameters to multi-objective performance is often complex and the Pareto front may be a non-connected manifold or a set of manifolds. Weighted aggregation of policies independently trained for distant scalarization weights or single objectives will produce suboptimal or even dominated policies that do not lie on the Pareto front. Consequently, rewarded soups can only provide a sparse set of discrete solutions and does not reconstruct the Pareto front manifold.
> In contrast, LLE-MORL explicitly learns and exploits local structure: by performing directional retraining and locally linear extension, it efficiently explores regions between solutions while remaining close to the true Pareto manifold. This enables a dense and efficient approximation of the Pareto front, which decomposition-based weighted aggregation methods cannot achieve.
>
> ---
> ## Clarification for the experimental evaluation ##
> MORL are indeed sensitive to hyperparameters. For this reason, we deliberately use standard, published configurations and apply them uniformly across all methods, rather than doing specific tuning only for our approach.
>
> Our experimental protocol follows standard practice in MORL: we use fixed, previously published hyperparameter configurations for both our method and the baselines, and we enforce equal environment-interaction budgets across all methods. Below, we clarify the setup in detail to avoid misunderstandings.
>
> - Hyperparameter for LLE-MORL and baselines
>
> For all comparisons in the main body, all hyperparameters of LLE-MORL are fixed according to Tables 4 and 5 (Appendix C.5); we do not change them across environments or settings. Appendix F is an additional sensitivity analysis, and these experiments explicitly show that our default values are not the best choices. Those typical and general parameters are chosen for simplicity and preliminary results. The additional results in Appendix F are used to study robustness, not to pick tuned hyperparameters.
>
> For the baselines, because the original papers do not evaluate on our exact benchmark problems, we follow the implementations and hyperparameter settings provided by morl-baselines [1]. Since LLE-MORL is PPO-based, we adopt PPO configurations recommended in prior PPO-based MORL work, in particular PGMORL [2], which uses commonly accepted PPO hyperparameters for continuous-control MORL tasks. Thus, both our method and the baselines rely on standard, previously published hyperparameter choices rather than environment-specific tuning.
>
> - Clarification of “to convergence” and Figure 3
>
> In Figure 3, “running PPO to convergence” means training the scalarised PPO policy under a fixed preference until the training curve plateaus within the given interaction budget. In continuous-control RL with deep networks, plateaued training does not imply that the resulting policy is globally optimal or Pareto-optimal, because local optima and finite budgets are standard limitations. Figure 3(c) illustrates exactly this: the “original policy” obtained by standard PPO can be Pareto-suboptimal, and our PPR-based locally linear extension then moves the solution into a better, Pareto-optimal region of the front. The existence of a dominating solution in that figure is precisely the motivation for adding our LLE stage on top of otherwise standard scalarised training, which is a promising non-training-based method which can efficiently explore the Pareto front manifold.
>
> In summary, our experiments use fixed, standard hyperparameters for both LLE-MORL and baselines, and all methods are evaluated under the same total environment interaction budgets. The improvements shown are therefore due to the algorithmic contributions of LLE-MORL (PPR + LLE) rather than to extra hyperparameter tuning.
>
> [1] Felten et al., “A toolkit for reliable benchmarking and research in multi-objective reinforcement learning,” NeurIPS 2023.
>
> [2] Xu et al., “Prediction-guided multi-objective reinforcement learning for continuous robot control,” ICML 2020.

---

### Official Review · Reviewer_eZe9 · 2025-10-30

**Soundness:** 1
**Presentation:** 2
**Contribution:** 2
**Rating:** 2
**Confidence:** 4

**Summary:**

This paper proposes a multi-objective reinforcement learning (MORL) algorithm for approximating a Pareto front of conflicting reward functions (objectives). The paper discusses the relationship between the performance space of a policy (its multi-objective expected return) and the parameter space of a policy e.g., the weights of a neural network). Then, the authors propose a population-based MORL method that generates novel policies via Locally Linear Extension (LLE). That is, for each policy, the method computes a vector of the differences between the policy network weights before and after being trained for a different preference over objectives. Next, the method generates novel policies by simply adding this parameter change to the original network weights using different scale factors. All policies generated using this process are then filtered, retaining only the non-dominated solutions. The method is evaluated and compared with existing MORL methods in the multi-objective MuJoCo benchmark from MO-Gymnasium.

**Strengths:**

- The problem of approximating a Pareto front of policies in RL is of interest to the community, and methods that can tackle this problem efficiently are highly relevant.
- The proposed method is simple to implement and computationally efficient.

**Weaknesses:**

- The paper misses discussing two very related and similar methods in the MORL literature (see [1] and [2] below). The work in [1] also explores the relationship between the performance and parameter spaces of Pareto optimal policies. However, instead of simply interpolating neural network weights, they train a hypernet to output the parameters of new policies given a preference vector. The general idea of their work is quite similar, and I would expect that their hypernet produces better results than the author’s proposed interpolation method, due to its higher expressiveness.
Moreover, the work in [2] also proposes to interpolate the weights of multiple neural network policies to generate other Pareto undominated policies efficiently (although in the RLHF setting).

- The proposed method has 8 different parameters that need to be tuned (see Table 5 in Appendix C.5). It seems these parameters were tuned during the sensitivity analysis in Appendix F. However, it is not detailed how the hyperparameters of the competing methods were tuned. It is not possible to infer whether the results are due to misrepresented baselines/lack of hyperparameter tuning, or due to the algorithmic contributions.
- The authors state they used multiple random seeds. Please state how many random seeds and what the dispersion metric was used in the results in the tables (standard deviation, standard error, etc).
- It is unclear whether the method can scale or would be easily applicable to domains with more than $d=2$ objectives.

[1] Shu, Tianye, Ke Shang, Cheng Gong, Yang Nan, and Hisao Ishibuchi. ‘Learning Pareto Set for Multi-Objective Continuous Robot Control’. Proceedings of the Thirty-Third International Joint Conference on Artificial Intelligence, 2024.

[2] Rame, Alexandre, Guillaume Couairon, Corentin Dancette, et al. ‘Rewarded Soups: Towards Pareto-Optimal Alignment by Interpolating Weights Fine-Tuned on Diverse Rewards’. Thirty-seventh Conference on Neural Information Processing Systems, 2023.

**Questions:**

Below, I have some suggestions and questions for the authors:

- Since it is assumed a linear scalarization function, it is useful to define a convex coverage set (CCS) as the set of optimal policies in the convex region of the Pareto front in Section 2.2. It is known that you can not identify points in concave regions of the Pareto front using linear scalarization.

- The quality of the plots in Figure 2 is very low. The authors should re-generate them with higher resolution.

- In Definition 2, the Pareto front is defined using the letter $\mathcal{F}$, but in Section 3.5, the authors use the letter $P$.

- The assumptions A1 to A4 should be motivated and justified. For instance, I do not think it is trivial to guarantee the validity of A4.

- Hypervolume (HV) and Expected Utility (EU) -> The acronyms are being redefined multiple times in the text.

---

> ### Author Response · Authors · 2025-12-02
>
> Thank you for the review and for the opportunity to provide some clarification for our work.
>
> ## W1 - Related Work ##
>
> - ### Hyper-MORL [1] ###
> While previous work[1] proposes a manifold-based approach, its focus is different from ours. Their approach trains a hypernetwork that maps a preference vector to policy parameters in a learned low-dimensional subspace, and this hypernetwork is optimised end-to-end with PPO.
> However, [1] does not explicitly analyse or exploit how changes in policy parameters relate to changes in performance. The relationship between parameter space and performance space remains implicit: any structure is a by-product of end-to-end hypernetwork training, and is not used algorithmically. The authors only suggest that understanding this relationship could be interesting as future work.
>
> In contrast, LLE-MORL is built exactly around this connection. We formalise a local Parameter–Performance Relationship (PPR), empirically validate it, and then explicitly use it to design a concrete mechanism: directional retraining and locally linear extension in parameter space to generate new candidate policies without retraining each one from scratch. This leads to efficient and high-quality Pareto-front reconstruction from a small set of standard scalarised policies under a limited interaction budget.
>
> Regarding the reviewer’s concern that the method in [1] might perform better than ours, we note that their main baseline is PGMORL/PDMORL. We agree that PGMORL is closely related to their method, but it is not state-of-the-art on our benchmark suite. Moreover, if you check the experimental settings in [1], you will notice that they rely on very large numbers of environment interaction steps (hundreds or thousands times than our settings), which is typical for such MO-PPO style methods and where these approaches usually suffer from sample-inefficiency issues, just as we discussed in the related work section. Our LLE stage, by contrast, is a training-free extension step: once a small amount of base and directional policies have been obtained, generating additional candidate policies does not require further environment interaction. LLE-MORL is therefore specifically designed to provide efficient and high-quality Pareto-front reconstruction, especially under a limited interaction budget.
>
> Moreover, LLE-MORL is a modular, plug-and-play component rather than a new single-model architecture: the LLE stage can be applied on top of existing MORL backbones. Thus, while [1] introduces a new hypernetwork-based learner, our contribution is novel: we operationalise the parameter-performance relationship itself to reconstruct the Pareto front efficiently from existing policies.
>
>
> - ### Rewarded Soups [2] ###
> While rewarded soups and LLE-MORL both manipulate policy parameters, our approach provides a fundamentally different capability that those weighted aggregation methods cannot offer, the ability to locally capture and efficiently explore the Pareto front manifold.
>
> Rewarded soups is a decomposition-based method which utilises weighted aggregation approaches between multiple trained single-objective policies. This can combine discrete solutions but cannot directly explore or reconstruct the local Pareto front manifold between these solutions.
> This limitation arises because each policy is independently optimised for a different single-objective, it naturally lacks information about how small changes in policy parameters affect performance across multiple objectives, which is crucial for accurately tracing the local manifold of the Pareto front.
>
> Moreover, the mapping from policy parameters to multi-objective performance is often complex and the Pareto front may be a non-connected manifold or a set of manifolds. Weighted aggregation of policies independently trained for distant scalarization weights or single objectives will produce suboptimal or even dominated policies that do not lie on the Pareto front. Consequently, rewarded soups can only provide a sparse set of discrete solutions and does not reconstruct the Pareto front manifold.
> In contrast, LLE-MORL explicitly learns and exploits local structure: by performing directional retraining and locally linear extension, it efficiently explores regions between solutions while remaining close to the true Pareto manifold. This enables a dense and efficient approximation of the Pareto front, which decomposition-based weighted aggregation methods cannot achieve.
>
> [1] Shu, Tianye, Ke Shang, Cheng Gong, Yang Nan, and Hisao Ishibuchi. ‘Learning Pareto Set for Multi-Objective Continuous Robot Control’. Proceedings of the Thirty-Third International Joint Conference on Artificial Intelligence, 2024.
>
> [2] Rame, Alexandre, Guillaume Couairon, Corentin Dancette, et al. ‘Rewarded Soups: Towards Pareto-Optimal Alignment by Interpolating Weights Fine-Tuned on Diverse Rewards’. Thirty-seventh Conference on Neural Information Processing Systems, 2023.

---

> > ### Author Response · Authors · 2025-12-02
> >
> > ## W2 - LLE-MORL and baseline hyperparameters ##
> >
> > Appendix F is not related to the baselines or to how we select hyperparameters for the main comparisons. Its sole purpose is to illustrate how certain hyperparameters influence LLE-MORL’s behaviour, and to make clear that we do not choose hyperparameters to maximise performance on each settings. We also clarify that, for all baseline comparisons reported in the main body of the paper, the parameters for LLE-MORL strictly follow Appendix C.5, and we do not perform any additional hyperparameter tuning for our method.
> >
> > In Appendix F.1 (Effect of directional retraining shift $\delta_s$), we explicitly state (lines 1055–1060) that while a moderate retraining shift (e.g., $\delta_s$ in the range of 0.2 to 0.3) often provides a good balance between HV/EU and SP, our default configuration uses $\delta_s=0.1$, chosen for simplicity and promising preliminary results. This ablation demonstrates that the default shift we used in paper's main part is not universally optimal.
> >
> > As for the step-scale factors in locally linear extension, the default settings used for all comparisons (Table 5) employ a single, typical choice of $(\alpha_{start}, \alpha_{end}, \Delta\alpha)$ that generally balances performance and computational cost. We do not change these parameters across different settings or tasks for comparison. Table 8 shows that other choices can have better results, confirming again that our default is not tuned for best performance but a typical and general choice. Since this stage is not a training-based process, users can easily adjust these parameters in practice according to their own performance-efficiency trade-offs.
> >
> > Again, for all comparisons reported in the paper, all hyperparameters of LLE-MORL are fixed according to Tables 4 and 5 shown in Appendix C.5.
> > For the baselines, because the original papers do not evaluate on our exact benchmark problems, we follow the implementations and hyperparameter settings provided by morl-baselines [1]. Since LLE-MORL is PPO-based, we adopt PPO configurations recommended in prior PPO-based MORL work, in particular PGMORL [2], which uses commonly accepted PPO hyperparameters for continuous-control MORL tasks. Thus, both our method and the baselines rely on standard, previously published hyperparameter choices rather than environment-specific tuning.
> >
> > [3] Felten et al., “A toolkit for reliable benchmarking and research in multi-objective reinforcement learning,” NeurIPS 2023.
> >
> > [4] Xu et al., “Prediction-guided multi-objective reinforcement learning for continuous robot control,” ICML 2020.
> >
> > ---
> > ## W3 - Random seeds and dispersion metrics ##
> >
> > We thank the reviewer for pointing this out. 5 random seeds are used, and the values in the tables correspond to the mean and standard deviation across these runs.
> >
> > ---
> > ## W4 -  Scalability ##
> >
> > Our method is not restricted to 2 objectives. The algorithm is formulated for a general n-objective setting, as discussed in Appendix B.2:
> >
> > In an $n$-objective reinforcement learning problem, if the Pareto front in the policy parameter space can be represented as a differentiable $(n-1)$-dimensional manifold, LLE-MORL can locally reconstruct this manifold by extending each base policy along $(n-1)$ distinct directions obtained via directional retraining.
> > The locally linear extension then combines these directions to generate a grid of new candidate policies. In a $n$-objective task, $n-1$ directional retrained directions are used to form a $(n-1)$-dimensional grid: $\theta=\theta_{\rm base}+\sum\alpha_i(\theta_{i}-\theta_{\rm base})$, $i=1,\dots,n-1$, allowing dense coverage of the local Pareto surface with or without further training.
> >
> > Given that the number of objectives is $n$, the number of base policies is $K$, the base policy training time is $T$, the number of locally linear extensions sampled per direction is $M$, and noting that the locally linear extension is training-free, the expected running time of LLE-MORL is $O(TK+KM^{n-1})$.
> >
> > We have tested LLE-MORL on 3-objective settings, and in practice, the behaviour is consistent with this description as the theoretical analysis is straightforward and does not require any additional algorithmic changes.

---

### Official Review · Reviewer_T2kU · 2025-11-04

**Soundness:** 3
**Presentation:** 2
**Contribution:** 2
**Rating:** 4
**Confidence:** 4

**Summary:**

The paper introduces a novel approach for probing the Pareto-front of multi-objective reinforcement learning problems. The key idea of this approach is to take a pair of near-optimal policies parametrized by similar parameter vectors and evaluate a sequence of "intermediate policies" to map the appropriate piece of Pareto-front. Intermediate policies are obtained by sampling parameter vectors on the linear interval between the two original policies in the parameter space.

**Strengths:**

Novel approach for probing the Pareto front in MORL.

**Weaknesses:**

W1) The notion of Parameter-Performance Relationship (PPR) the authors introduce seems to basically require linearity of V in the region U, which is unlikely to hold for any meaningful region in general. The notion of local PPR is then used in the text without being explicitly defined. Presumably, local PPR is just differentiability in a particular point.

W2) The proposed theorem feels too vague to be useful. The meaning of "reconstructs this part of the Pareto front up to a resolution" is not clear.

W3) The meaningfulness of the comparisons presented in evaluation is not clear to me. Different approaches produce different numbers of points in the performance space. Different approaches seem to have different computational budgets. The only thing that is similar between the approaches in the two presented settings is the "number of training steps". However, this number only controls training a single base policy, while any re-training steps or the number of base policies are
not controlled by this parameter.

**Questions:**

Q1) There are several mentions of "structurally similar policies" throughout the text, but the notion is never defined. What structure is implied here and why is it similar between the mentioned pairs of policies?

Q2) Figures 4 and 5 seem to indicate that the base policies themselves perform much better than the competition. Could the overall performance improvement then be attributed to simply having better training routine than what's used in the other approaches? I would love to see an additional ablation experiment, where the original 6 base policies are output as is with no further modifications. Such an experiment would help determine how performant the LLE process is by itself.

---

> ### Author Response · Authors · 2025-12-02
>
> Thank you for the review and for the opportunity to provide some clarification for our work.
>
> ---
> ## W1 - Definition of PPR ##
>
> Our Definition 3 of the Parameter-Performance Relationship (PPR) does not assume global linearity of $V$ over a large, “meaningful” region, but formalises an idealised local structure that we then use in an approximate sense in neighbourhoods around the base policies.
> What we mean by local PPR is precisely this neighbourhood-level relationship between directional parameter updates and the induced directional shifts in the performance space.
> As the comparison with other methods show this is fully sufficient to find very efficiently near-Pareto fronts that are strikingly competitive when compared with other methods. If the true Pareto front is meant to be found, which seems to be what this referee might have in mind, then it is possible to run for each point on the Pareto front a further detailed approximation, but as no convergence proofs for optimality in Deep RL exist, it is not possible to prove anything about this.
>
> In a revised version, we will (i) state clearly that PPR is meant as a local, approximate structural assumption rather than a global linearity condition, and (ii) explicitly define “local PPR” as the restriction of Definition 3 to neighbourhoods around the policies considered, where it is empirically supported by the retraining behaviour.
>
> ---
> ## W2 -  Theorem ##
>
> The theorem provides a statement on the generality of our algorithm, which finds its limits in fractal Pareto fronts or discrete fronts with convergence points, even if the base points are already optima, as specified by our theorem.
> We are not aiming at a full characterisation of mathematically possible Pareto fronts, which would naturally include a number of practically irrelevant problems.
>
> The problem of resolution is addressed by the complexity result, which is similar to earlier work on algorithms that, like our approach, aim at an analysis of non-trivial compromises among the objectives, whereas better results are possible only if the objectives enter the optimisation process via a simple superposition.
>
> ---
> ## W3 - Evaluation fairness ##
>
> It is expected in MORL that different approaches produce different numbers of final points on the Pareto front. Our comparisons are made on the resulting nondominated sets under the same total training budget. We are talking here about reinforcement learning, where the critical factor is the environment interaction.
>
> In all experiments, we fix a global training budget in environment steps for each environment and enforce that every method, including LLE-MORL, uses at most this budget. For LLE-MORL, this budget includes: training all base policies, all directional retraining steps, and all fine-tuning steps.
> Thus, the “number of training steps” does not only refer to training a single base policy; retraining and fine-tuning are fully counted inside the same total budget.
> If we vary the number of base policies in LLE-MORL, the per-policy training steps will be automatically adjusted so that the overall training budget remains fixed.
>
>
> ---
> ## Q1 -  Structurally similar policies ##
>
> We thank the reviewer for highlighting this point, which is central to the main idea of our approach.
> By “structurally similar policies”, we refer to policies that share the same network architecture and differ only by small, structured changes in their parameters induced by short retraining under different preference vectors.
> In our setting, all policies use the same neural network topology. Changing the preference $\omega$ only affects the learned weights.
>
> ---

---

> > ### Author Response · Authors · 2025-12-02
> >
> > ---
> > ## Q2 - The role of base policies and the LLE process ##
> >
> > Figures 4 and 5 show the final set returned by LLE-MORL, i.e., the non-dominated set of all extrapolated and fine-tuned policies. They do not indicate that the base policies alone are substantially better than the competition.
> > As explicitly illustrated in Figure 3, the initial base policy could be Pareto-suboptimal, and the locally linear extension using PPR moves the solution into a Pareto-optimal region of the front.
> > The base policies themselves are standard multi-objective PPO networks. As discussed in the related work section, such scalarised MO-PPO training is not sufficient to reconstruct a high-quality Pareto front, especially under the limited interaction budget.
> >
> > The LLE process connects the learning dynamics in the parameter space with the evaluation in the performance space. It does not use any fitness evaluations, so in the sense of RL, it cannot be performant by itself. The value of our approach lies in the observation that the results produced by the LLE process do not appear to lose much of the initially injected performance while substantially exploring the Pareto front manifold and improving the coverage. Obviously, without a reasonable base policy performance, nothing can be produced magically through the LLE approach.
> >
> > Our key contribution is not designing stronger base policies, but showing that, once a few standard policies are available, we can reconstruct a high-quality Pareto front efficiently by exploiting the Parameter-Performance Relationship (PPR) and our locally linear extension (LLE) mechanism. The value of LLE-MORL lies in this efficient and high-quality front reconstruction, not in the particular choice of backbone.
> >
> > Moreover, the main part of LLE-MORL is a plug-and-play module: the locally linear extension and Pareto front reconstruction stages can, in principle, be combined with different backbone learners, non-linear scalarisation techniques, or more sophisticated interpolation/extrapolation strategies. In this sense, our current base policies are deliberately kept typical PPO networks, precisely to highlight that the improvements come from the PPR-based reconstruction mechanism, rather than from a specially engineered base-training routine.

---

### Official Review · Reviewer_xqCc · 2025-11-05

**Soundness:** 3
**Presentation:** 2
**Contribution:** 3
**Rating:** 6
**Confidence:** 4

**Summary:**

This paper proposes LLE-MORL, a new method for multi-objective reinforcement learning (MORL) that leverages a locally linear parameter-performance relationship (PPR) between policies. The authors show that small, structured weight updates caused by short retraining under new preferences can be used to extrapolate additional candidate policies without full training. The key idea is to:
1.	Train several base policies under different preference vectors.
2.	Conduct short retraining with neighboring preferences to acquire directional parameter updates.
3.	Linearly extrapolate parameters along these directions to produce candidate policies.
4.	Perform short fine-tuning to refine each candidate back to the Pareto front.
Experiments across continuous-control MO-Gym environments demonstrate both sample-efficiency and improved Pareto-front quality.

**Strengths:**

Novel conceptual insight: The work introduces and formalizes the parameter-performance relationship (PPR) for MORL and empirically validates it, which is conceptually interesting and bridges representation geometry and RL optimization.
Ablation study demonstrates necessity of retraining vs. extension and the benefit of fine-tuning.
Theoretical characterization of when the method reconstructs the Pareto front.

**Weaknesses:**

Limited benchmark diversity: Only continuous control environments from MO-Gymnasium are used. Additional settings (e.g. discrete tasks, real-world robotics, offline MORL) would strengthen conclusions.
Computational trade-offs: While extrapolation is cheap, initial base policy training + retraining overhead may be non-trivial. A fairer wall-clock runtime comparison table would be helpful.
Comparisons with related work on Pareto front analysis: A more thorough comparison against some similar Pareto front analysis for MORL.
Writing style: The paper is written in a dense form and requires a thorough reading, there are references that are missing names of all authors or incomplete information. Please increase the font sizes of all text in all plots especially in Figures 3, 4 and 5. Also the numbers in the Tables are really small while the figures can be re-positioned so that it allows for additional space and increase the font sizes.

**Questions:**

1.	How sensitive is performance to \alpha step sizes and number of base policies?
Can adaptive \alpha or uncertainty-based step-control improve robustness?
2.	Have you tested higher-dimensional objectives (d > 2)?
Does efficiency degrade in practice following the theoretical M^{n-1}complexity?

---

> ### Author Response · Authors · 2025-11-30
>
> We sincerely thank your time and effort in reviewing our submission. In the following, we provide detailed responses and clarifications to the specific concerns raised about our work.
>
> ---
> ## Weakness ##
> Regarding the computational trade-offs, in all our experiments, all approaches are run under the same total training budget in environment timesteps, which ensures a fair comparison in terms of compute and sample efficiency. The execution time for each method and benchmark, under both the sample-efficient and standard training settings, is reported in Appendix C.6, allowing readers to directly inspect and compare wall-clock runtime across methods.
>
> We appreciate the reviewer’s suggestions on improving the writing.
>
> ---
> ## Q1: Sensitivity to $\alpha$ step sizes and number of base policies ##
>
> For the algorithm's sensitivity to the choice of step-scale factors $\alpha$ used during the locally linear extension process (Stage 3, Section 3.5), we clarify that $\alpha$ indeed has a meaningful but limited influence on the performance of the algorithm.
> As illustrated in Figure 3 and Appendix F.2, $\alpha$ controls how far we move along the locally linear direction between a base policy $\theta_{\omega}$ and its directional retrained policy $\theta_{\omega'}$:
> when $\alpha<1$, extension remains between $\theta_{\omega}$ and $\theta_{\omega'}$, preventing exploration of new regions of the Pareto Front beyond those two endpoints;
> when $\alpha$ is too large, the algorithm may step outside the local region where the Parameter-Performance Relationship (PPR) is valid, and further extrapolation typically does not yield improved solutions.
>
> Thus, there exists an effective range of $\alpha$ values (see Appendix F.2), where the PPR assumption holds and extension is effective.
> Within this range, LLE-MORL maintains stable performance in terms of both Hypervolume (HV) and Expected Utility (EU). Regarding the Sparsity (SP) metric, its value is mainly influenced by the number and interval of $\alpha$ values used during extension, which controls the density of points along the Pareto front, rather than by the exact value of a single $\alpha$.
> Our experiments demonstrate that this effective range of step-scale factors $\alpha$ is not too wide and is easy to control, enabling a good balance between performance and efficiency. Furthermore, since $\alpha$ values beyond the effective range do not degrade the already found solutions, the algorithm remains robust in practice.
>
> For the number of base policies, using a moderate set of base preferences already provides good coverage, and additional base policies mainly densify the front with diminishing returns relative to their extra cost. In all reported experiments we use the same number of base policies across tasks, without per-environment tuning, which suggests that LLE-MORL is not overly sensitive to this choice. Finally, adaptive $\alpha$ schemes or uncertainty-based step control (e.g., selecting $\alpha$ based on validation performance or local uncertainty) are natural extensions of our framework; we leave such adaptive strategies for future work, as the simple fixed $\alpha$-grid used here already yields robust and competitive performance.
>
> ---
> ## Q2: Higher-dimensional Objectives ##
>
> We have tested LLE-MORL on 3-objective settings, and in practice, the behaviour is consistent with this theoretical description, as the theoretical analysis is straightforward and does not require any additional algorithmic changes.

---

### Note · Authors · 2026-01-22

I have read and agree with the venue's withdrawal policy on behalf of myself and my co-authors.